# Modular and Adaptive Conformal Prediction for Sequential Models via Residual Decomposition

## Abstract

Conformal prediction offers finite-sample coverage guarantees under minimal assumptions. However, existing methods treat the entire modeling process as a black box, overlooking opportunities to exploit modular structure. We introduce a conformal prediction framework for two-stage sequential models, where an upstream predictor generates intermediate representations for a downstream model. By decomposing the overall prediction residual into stage-specific components, our method enables practitioners to attribute uncertainty to specific pipeline stages. We develop a risk-controlled parameter selection procedure using family-wise error rate (FWER) control to calibrate stage-wise scaling parameters, and propose an adaptive extension for non-stationary settings that preserves long-run coverage guarantees. Experiments on synthetic distribution shifts, as well as real-world supply chain and stock market data, demonstrate that our approach improves coverage under stage-wise shifts compared to standard conformal methods, while providing identification of stage-wise error contribution. This framework offers diagnostic advantages and robust coverage that standard conformal methods lack.

## 1 Introduction

Modern machine learning systems increasingly rely on modular pipelines, where upstream predictors generate intermediate representations for downstream models. Such pipelines appear across diverse applications: macroeconomic forecasting, where supply chain indicators inform market predictions, and medical diagnosis, where imaging features guide treatment decisions (Crane and Crotty, 1967; Soybilgen and Yazgan, 2021). In these settings, uncertainty quantification is essential and should reflect how error propagates across stages. Conformal prediction (Vovk et al., 2005) provides principled prediction intervals with finite-sample coverage guarantees under exchangeability. However, existing conformal prediction methods treat these multi-stage systems as monolithic black boxes, overlooking their modular structure and missing opportunities for targeted error attribution, i.e. identification of the major source of error.

Recent work expands the flexibility of conformal prediction through reweighting and adaptive calibration (Barber et al., 2023; Angelopoulos et al., 2023; Oliveira et al., 2024; Farinhas et al., 2023; Angelopoulos et al., 2024; Gibbs and Candès, 2021; 2024), structured model-aware adaptations (Zargarbashi et al., 2023; Zhang et al., 2025; Wu et al., 2025), and risk-control (Bates et al., 2021; Angelopoulos et al., 2021), but largely focuses on single-stage or black-box models, rather than taking a modular perspective.

This perspective is especially pertinent under distribution shift, which often affects stages asymmetrically—e.g., upstream sensors may drift while downstream mappings remain stable. Standard conformal methods cannot disentangle such effects, which may force practitioners to retrain the entire pipeline when targeted interventions would suffice. To address this gap, we propose a stage-wise abstraction that isolates these effects, embedding stage-wise uncertainty into conformal prediction, yielding robust, intervals under distribution shift that quantify the error produced at each stage.

By decomposing residuals into stage-wise components, our method constructs prediction intervals through a linear combination of stage-specific quantiles, weighted by scaling parameters selected via family-wise error rate (FWER) control over a calibration set. This yields valid intervals without

requiring internal model access—only structural knowledge of the pipeline. Our approach offers two key advantages over existing conformal methods: (i) it provides diagnostic transparency, identifying which stages contribute most to uncertainty, and (ii) it improves coverage under distribution shifts affecting individual components where standard approaches degrade.

For intuition, we briefly introduce the two-stage setting with triplets $(w, x, y)$ and latent structure $x = \mu_1(w) + \varepsilon_1$, $y = \mu_2(x) + \varepsilon_2$, where $\varepsilon_1, \varepsilon_2$ denote additive noise, and the model learns estimators $\hat{\mu}_1$, $\hat{\mu}_2$. For example, in automobile supply chains, $w$ could denote semiconductor prices, with $\hat{\mu}_1(w)$ estimating new vehicle demand $(x)$, and $\hat{\mu}_2(x)$ predicting used vehicle prices $(y)$. Our method decomposes the residual $R = |y - \hat{\mu}_2(\hat{\mu}_1(w))|$ into upstream $(\hat{\mu}_1)$ error component $\Delta R_1$ and downstream $(\hat{\mu}_2)$ residual $R_2$, capturing the uncertainty of each stage.

We also extend our framework to adaptive settings for which we update scaling parameters based on component-wise empirical coverage, improving responsiveness to (i) upstream, (ii) downstream, and (iii) end-to-end distribution shifts. We illustrate our method on synthetic shifts and real-world supply chain and financial data, demonstrating the ability to explicitly adapt to stage-wise shifts, and showing improved robustness over adaptive conformal baselines (Gibbs and Candès, 2021; 2024; Angelopoulos et al., 2023; 2024). For simplicity, we focus on two-stage models, but our methodology can be extended to multi-stage models (Appendix C).

**Contributions:**

- We propose a residual decomposition framework for sequential multi-stage models, that partitions prediction error into distinct upstream and downstream components, enabling stage-wise uncertainty attribution (Section 3).

- We develop a risk-controlled parameter selection procedure using FWER-based hypothesis testing to construct valid prediction intervals from decomposed residuals with coverage guarantees (Section 5).

- We introduce an adaptive algorithm that dynamically adjusts scaling parameters based on component-wise coverage feedback, preserving long-run coverage guarantees while providing stage-sensitive diagnostics for distribution shifts (Section 6).

- We demonstrate the framework's effectiveness on synthetic shifts and real-world economic forecasting, showing maintained coverage under conditions that degrade existing conformal methods, with diagnostic capabilities that enable targeted model interventions (Section 7).

**Paper Outline.** In Sections 2 and 3, we formalize the two-stage prediction setting and introduce our residual decomposition. Section 4 describes our method for constructing stage-aware prediction intervals, followed by the FWER calibration procedure in Section 5. We extend our method to an adaptive version Section 6, and evaluate performance under distribution shift in Section 7.

## 2 PROBLEM SETTING AND ASSUMPTIONS

We consider a sequential two-stage prediction problem. Each data point is characterized by a triplet $z = (w, x, y)$ where $w \in \mathcal{W}$ is the input to the first-stage model (upstream features), $x \in \mathcal{X}$ is an intermediate representation, and $y \in \mathcal{Y}$ is the final prediction target. We learn predictors $\hat{\mu}_1 : \mathcal{W} \to \mathcal{X}$ and $\hat{\mu}_2 : \mathcal{X} \to \mathcal{Y}$ which are composed to form end-to-end predictor $\hat{\mu}_2(\hat{\mu}_1(w)) = \hat{\mu}_2(\hat{x})$, where $\hat{x} = \hat{\mu}_1(w)$.

To learn these models and perform conformal prediction, we assume access to three disjoint subsets: (i) a *training* set $S^{\text{train}} = \{(w_i, x_i, y_i)\}_{i=1}^{n}$ used to fit both stages of the model via $(w, x)$ and $(x, y)$ pairs for each stage respectively; (ii) a *conformal* set $S^{\text{conf}} = \{(w_i, x_i, y_i)\}_{i=n+1}^{n+m}$ for computing non-conformity scores; and (iii) a *calibration* set $S^{\text{cal}} = \{(w_i, x_i, y_i)\}_{i=n+m+1}^{n+m+l}$ for parameter selection. At prediction time, only the upstream input $w_{\text{test}}$ is observed, while the intermediate value $x_{\text{test}}$ and target $y_{\text{test}}$ are unobserved and must be predicted.

We list some assumptions that we consider at different points throughout the paper. (i) **Exchangeability.** Unless stated otherwise, for theoretical guarantees, we assume the data $(w, x, y)$ in $S^{\text{train}}, S^{\text{conf}}, S^{\text{cal}}$ to be exchangeable, as well as the test point $z_{\text{test}} = (w_{\text{test}}, x_{\text{test}}, y_{\text{test}})$. (ii) **Learning algorithms.** We assume that the algorithms that learn $\hat{\mu}_1, \hat{\mu}_2$ are deterministic, i.e. given the same

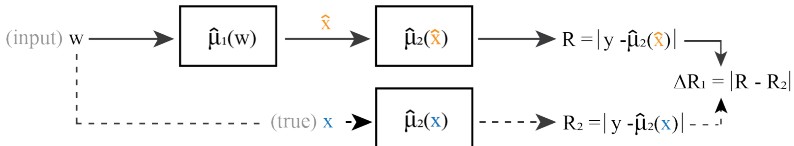

Figure 1: Visualization of a sequential two-stage model with residual components $R, \Delta R_1, R_2$.

training data, they produce identical predictors; and we also assume they are symmetric, i.e. invariant to permutations of the data. We also assume that the intermediate variable $x$ is observable for the given datasets, enabling residual decomposition. (iii) **Distribution shifts.** Distribution shifts violate exchangeability and can impact each prediction stage differently. Let $P$ denote the distribution of a data point in the training set and $P'$ denote the distribution of the test point. We consider three types of shift: *Upstream covariate shift*: $P(w) \neq P'(w)$; *Upstream (concept) shift*: $P(x|w) \neq P'(x|w)$; and *Downstream (concept) shift*: $P(y|x) \neq P'(y|x)$.

**Objective.** Given test upstream input $w_{\text{test}}$, the goal is to construct prediction interval $\hat{C}_\alpha(w_{\text{test}})$ such that $\mathbb{P}(y_{\text{test}} \in \hat{C}_\alpha(w_{\text{test}})) \geq 1 - \alpha$, where $\alpha \in (0, 1)$ is the target miscoverage level. Under exchangeability, this is the standard conformal prediction objective. However, under distribution shifts, we seek to maintain robust coverage while providing identification of the impact of specific pipeline stages on prediction uncertainty.

The key challenges in this setting are: (i) **Attribution**: Understanding which stage contributes more to prediction uncertainty; (ii) **Adaptivity**: Maintaining coverage under shifts affecting different stages; (iii) **Transparency**: Providing actionable insights for model improvement or retraining decisions. Our approach addresses these challenges by decomposing the end-to-end prediction error into stage-specific components for targeted uncertainty quantification and robust interval construction. While we define these concepts for two-stage models, we discuss auxiliary inputs, multiple upstream models, and deeper sequential pipelines in Appendix C. We also provide a notation table in Appendix B.2.

## 3 TWO-STAGE RESIDUAL DECOMPOSITION FOR CONFORMAL PREDICTION

To address the aforementioned challenge of attribution, we partition the total prediction residual $R(w, y) = |y - \hat{\mu}_2(\hat{\mu}_1(w))|$ into upstream and downstream components. This decomposition enables stage-wise attribution of error, in contrast to standard black-box conformal methods. We provide a visualization in Figure 1.

**Definition 1** (Second-stage residual). *Given a point $(w, x, y)$ and downstream predictor $\hat{\mu}_2 : \mathcal{X} \to \mathcal{Y}$, the second-stage residual is*
$$R_2(x, y) = |y - \hat{\mu}_2(x)|.$$
*This captures downstream prediction error when given the true intermediate input $x$.*

**Definition 2** (First-stage delta). *Let $\hat{x} = \hat{\mu}_1(w)$ be the output of the first-stage predictor. The first-stage delta is defined as*
$$\Delta R_1(w, x, y) = | |y - \hat{\mu}_2(x)| - |y - \hat{\mu}_2(\hat{x})| |.$$
*This quantifies the change in downstream prediction error induced by replacing the true intermediate $x$ with its prediction $\hat{x}$.*

For notational purposes, we drop the dependence on $(w, x, y)$, and refer to these terms as $R, \Delta R_1$, and $R_2$. Intuitively, $\Delta R_1$ reflects the error of the upstream predictor, while $R_2$ isolates downstream error without upstream influence. This decomposition satisfies a fundamental upper-bound property.

**Proposition 1.** *For any point $(w, x, y)$, the total residual satisfies:*
$$R = |y - \hat{\mu}_2(\hat{\mu}_1(w))| \leq \Delta R_1 + R_2.$$

By definition, $\Delta R_1 = ||y - \hat{\mu}_2(x)| - |y - \hat{\mu}_2(\hat{x})||$. Let $A = |y - \hat{\mu}_2(x)| = R_2$ and $B = |y - \hat{\mu}_2(\hat{x})| = R$. By the reverse triangle inequality, $B \leq A + |A - B| = R_2 + \Delta R_1$. This upper bound property

ensures that the sum of components provides a conservative estimate of the total error, which is crucial for the coverage guarantees developed subsequently.

This decomposition provides a clean interpretation: $R_2$ measures the inherent uncertainty of the downstream model, while $\Delta R_1$ measures how upstream prediction errors affect the final prediction error magnitude. When $\Delta R_1$ is small relative to $R_2$, the upstream predictor performs well and downstream uncertainty dominates. Conversely, when $\Delta R_1$ is large relative to $R_2$, upstream errors drive prediction uncertainty. This attribution enables practitioners to identify which stage requires improvement and guide retraining decisions. Furthermore, these components provide insights for handling distribution shifts: under upstream shifts, $\Delta R_1$ typically increases as the first-stage predictor encounters out-of-distribution inputs, while under downstream shifts, $R_2$ increases. These changes can be experimentally visualized in Appendix Figure 14b. Thus, the varied responses enable targeted adaptive strategies for different distribution shifts.

These components exhibit intuitive relationships with the total residual: when $R_2$ is small, $\Delta R_1$ closely approximates $R$, while when the upstream model is accurate and $\hat{\mu}_2$ is smooth, $R_2$ closely approximates $R$ (see Appendix A.2). Importantly, at least one component must represent a majority of the error, ensuring meaningful stage-wise attribution. Next, we describe two complementary approaches to combining these residual components into prediction intervals.

## 4 CONSTRUCTING PREDICTION INTERVALS

We describe two approaches that incorporate $\Delta R_1, R_2$, utilizing the conformal set $S^{\text{conf}}$, to compute component-wise quantiles, with data-driven parameter selection using $S^{\text{cal}}$ addressed in Section 5. Note that there exists a rich space of heuristics for combining these residual components beyond those listed below—see Appendix C.1. For proofs of the following results, see Appendix A.1.

### 4.1 SEPARATE COMPONENT QUANTILES

We compute sets of residual components on $S^{\text{conf}}$: $\{\Delta R_1^i\}_{i=1}^m$ and $\{R_2^i\}_{i=1}^m$, which we abbreviate as $\{\Delta R_1\}$ and $\{R_2\}$. Prediction intervals are constructed by summing quantiles computed separately from each set, with the quantile levels controlled by $c$ and $d$ for the respective stages.

**Definition 3** (Separate component quantiles). *Let $c, d \in (0, 1)$ be quantile levels. For test input $w_{test}$, the prediction interval is constructed by summing separate quantiles of each component at $c, d$ levels:*

$$\hat{C}_{c,d}(w_{test}) = \hat{\mu}_2(\hat{\mu}_1(w_{test})) \pm (Q_{1-c}(\{\Delta R_1\}) + Q_{1-d}(\{R_2\})).$$

This construction inherits coverage guarantees from standard conformal prediction:

**Theorem 1** (Coverage of separate component quantiles). *Under the assumption of exchangeability, for $c, d \in (0, 1)$, the prediction interval $\hat{C}_{c,d}(w_{test})$ satisfies*

$$\mathbb{P}(y_{test} \in \hat{C}_{c,d}(w_{test})) \geq 1 - c - d.$$

### 4.2 SCALED COMPONENT QUANTILES

Our second approach fixes a quantile level $\alpha \in (0, 1)$ for both components and selects scaling parameters $a, b \in [0, 1]$ to weight their respective contributions. This provides understandable control over stage-wise attribution of uncertainty.

**Definition 4** (Scaled component quantiles). *For a fixed quantile $\alpha \in (0, 1)$ and scaling coefficients $a, b \in [0, 1]$, the prediction interval is*

$$\hat{C}_{\alpha,a,b}(w_{test}) = \hat{\mu}_2(\hat{\mu}_1(w_{test})) \pm (a \cdot Q_{1-\alpha}(\{\Delta R_1\}) + b \cdot Q_{1-\alpha}(\{R_2\}))$$

*where the quantiles are computed over $S^{conf}$.*

The choice of scaling parameters $a$ and $b$ provides clear control: setting $a = 0$ ignores upstream uncertainty while $b = 0$ focuses only on upstream effects. However, coverage guarantees for arbitrary choices of $(a, b)$ require careful analysis. For fixed weights, we have the following:

**Corollary 1** (Coverage with $a = b = 1$). *Under exchangeability, for $a = b = 1$ and $\alpha \in (0, 1)$, the interval $\hat{C}_{\alpha,a,b}(w_{test})$ satisfies*

$$\mathbb{P}(y_{test} \in \hat{C}_{\alpha,a,b}(w_{test})) \geq 1 - 2\alpha.$$

*Proof.* This follows directly from Theorem 1 with $c = d = \alpha$. □

For appropriately chosen quantiles and scaling weights, we observe that both methods can yield similar intervals. To provide maximum flexibility for both theoretical analysis and implementation, we can combine both approaches into a general framework:

$$\hat{C}_{a,b,c,d}(w_{\text{test}}) = \hat{\mu}_2(\hat{\mu}_1(w_{\text{test}})) \pm (a \cdot Q_{1-c}(\{\Delta R_1\}) + b \cdot Q_{1-d}(\{R_2\})),$$

This unified form allows independent control of both quantile levels and scaling parameters, enabling fine-tuned balance between coverage guarantees and stage-wise attribution of error. We provide guidance on default choices of $c, d$ in Appendix D.2 and discuss the choices of $(a, b)$ in the following sections.

### 4.3 COVERAGE FOR SCALED PARAMETERS

While coverage for intervals of the form $\hat{C}_{c,d}$ (Definition 3) follows from standard conformal analysis, establishing similar guarantees for intervals with scaled residuals $\hat{C}_{\alpha,a,b}$ (Definition 4) with arbitrary weights $(a, b) \in [0, 1]$ presents significant challenges. The scaling parameters $(a, b)$ have no direct mapping to quantile levels, making it difficult to derive explicit guarantees. Despite this, we can establish that valid scaling parameters exist in principle. Under mild regularity conditions, there always exist optimal scaling parameters $(a^*, b^*)$ that yield exact marginal coverage:

**Proposition 2** (Existence of ideal scaling parameters). *For desired coverage level $1 - \alpha$, $\exists a^*, b^* \in [0, 1]$ such that the interval with those scaling parameters satisfies the marginal coverage guarantee*

$$\mathbb{P}\left(y_{test} \in \hat{C}_{\alpha/2,a^*,b^*}(w_{test})\right) = 1 - \alpha,$$

*provided the distribution of the residual $R$ has no point masses.*

Thus for some desired coverage level $\alpha$, for residual component pieces taken at quantile level $1 - \alpha/2$, there exist (possibly many) scaling weights that provide exact coverage. However, finding optimal scaling parameters $(a^*, b^*)$ requires knowledge of the residual distribution, which is unavailable in practice. This creates a fundamental trade-off: scaled intervals offer direct stage-wise control of stage-wise error contribution but lack accessible guarantees, while separate quantile intervals provide coverage under minimal assumptions but offer less direct understanding of stage-wise error. To resolve this trade-off, we adopt a conservative risk-controlled approach (Bates et al., 2021; Angelopoulos et al., 2022; 2021) that selects parameters $(a, b)$ with demonstrable coverage $\geq 1 - \alpha$ over recent points, using $S^{\text{cal}}$, providing both theoretical guarantees as well as empirical robustness to the distribution shifts that motivate our framework. Section 5 formalizes this through a risk-controlled calibration procedure, and Section 6 extends it to adaptive parameter updates that respond to component-wise coverage feedback.

### 5 RISK-CONTROLLING APPROACH WITH RESIDUAL COMPONENTS

Since finding $(a^*, b^*)$ is impossible in practice, we reframe the problem as filtering for $(a, b)$ coefficient pairs that satisfy the nominal coverage level $1 - \alpha$. We select scaling parameters through multiple hypothesis testing with FWER control. For each $\lambda = (a, b) \in \Lambda$, we test whether miscoverage exceeds $\alpha$; FWER ensures we rarely accept poor parameters. This conservative approach provides three benefits: (i) robustness buffer under moderate shift, (ii) diagnostic abstention ($\Lambda_{val} = \emptyset$) signaling when retraining is needed, and (iii) flexible search over attributable error decompositions.

## 5.1 TESTING MISCOVERAGE VIA EMPIRICAL RISK

We aim to identify scaling parameters $\lambda = (a, b) \in [0, 1]^2$ for which the resulting prediction interval $\hat{C}_\lambda(w)$ achieves coverage at least $1 - \alpha$. Since theoretical guarantees for arbitrary $\lambda$ are unavailable, we perform a hypothesis test for whether its miscoverage rate exceeds $\alpha$. We fix a finite candidate set $\Lambda \subseteq [0, 1]^2$ of scaling pairs and define the corresponding prediction interval for each $\lambda = (a, b) \in \Lambda$:

$$\hat{C}_\lambda(w) = \{y : |y - \hat{\mu}_2(\hat{\mu}_1(w))| \leq a \cdot Q_{1-c}(\{\Delta R_1\}) + b \cdot Q_{1-d}(\{R_2\})\},$$

using the general framework introduced earlier that combines Definition 4 and Definition 3. Here, the quantile parameters $c$ and $d$ are fixed in advance with quantiles taken over $S^{\text{conf}}$, while the miscoverage testing is performed using $S^{\text{cal}}$ to identify suitable $\lambda$.

## 5.2 COMPUTING $p$-VALUES FROM CALIBRATION DATA

Let $l = |S^{\text{cal}}|$ denote the size of the calibration set. For each choice of scaling parameters $\lambda \in \Lambda$, we define the empirical miscoverage rate $\hat{\mathcal{R}}(\lambda) = \frac{1}{l} \sum_{i=1}^l \mathbb{1}_{\{y_i \notin \hat{C}_\lambda(w_i)\}}$. Under the stronger assumption that the calibration points are IID, and that the true miscoverage rate for a given $\lambda$ is constant, the number of missed points follows a Binomial distribution with parameters $l$ and the underlying miscoverage probability. For each $\lambda$, define the null hypothesis

$$\mathcal{H}_0(\lambda) : \mathbb{P}(y \notin \hat{C}_\lambda(w)) > \alpha.$$

Thus, for each $\lambda$ we define a $p$-value $p_\lambda = \mathbb{P}\left(\text{Bin}(l, \alpha) \leq l\hat{\mathcal{R}}(\lambda)\right)$. The $p$-values $p_\lambda$ are super-uniform under $\mathcal{H}_0$ (Appendix A.2); that is, for any $u \in [0, 1]$, we have $\mathbb{P}(p_\lambda \leq u) \leq u$, which is crucial for FWER-controlling guarantees (Bates et al., 2021). Thus, we apply FWER-controlling multiple testing algorithms (Appendix B.4) to the collection of $p_\lambda$ to obtain the set of valid scaling parameters $\Lambda_{\text{val}} \subseteq \Lambda$. This ensures that, with probability at least $1 - \delta$ for some $\delta > 0$, no $\lambda$ with a miscoverage rate exceeding $\alpha$ is accepted. Note that the $\lambda \in \Lambda_{\text{val}}$ are more conservative as they require evidence of coverage of *at least* $1 - \alpha$. In contrast, other prediction interval methods yield coverage that is *merely close* to $1 - \alpha$. This conservatism may result in unnecessarily wide intervals, particularly when coverage is less critical than efficiency, such as in IID settings. To remedy this, we can include tolerance parameter $\tau$ and calculate $p_\lambda$ with $\text{Bin}(l, \alpha + \tau)$, trading some guarantees for practical performance. As demonstrated in our experiments (Appendix D.3.2), even small values of $\tau$ significantly improve efficiency while maintaining empirical coverage under IID settings.

## 5.3 COVERAGE GUARANTEES VIA RISK-CONTROLLED SCALING

Thus, given the $p$-values $p_\lambda$ and a FWER-controlling multiple testing algorithm (Appendix B.4), we identify the set of validated scaling parameters $\Lambda_{\text{val}} \subseteq \Lambda$. The following result states any intervals associated with $\Lambda_{\text{val}}$ achieve coverage with high probability:

**Theorem 2** (Risk control via FWER calibration). *Let $\Lambda_{val} \subseteq \Lambda$ be the set selected by a FWER-controlling algorithm at level $\delta$, based on $p$-values computed over the calibration set $S^{cal}$ with tolerance $\tau > 0$. Then, for any $\hat{\lambda} \in \Lambda_{val}$, we have*

$$\mathbb{P}\left(\mathbb{P}\left(y_{test} \in \hat{C}_{\hat{\lambda}}(w_{test}) \,\middle|\, S^{cal}\right) \geq 1 - \alpha - \tau\right) \geq 1 - \delta,$$

*where the outer probability is over the randomness of $S^{cal}$, and the inner probability is over the test point $(w_{test}, x_{test}, y_{test})$.*

This guarantee follows from the FWER-controlling property: by reducing the probability of false positives, i.e. selecting scaling parameters with poor coverage, we ensure that all selected parameters satisfy the coverage requirement with high probability. Thus, with probability at least $1 - \delta$ over $S^{\text{cal}}$, any selected interval $\hat{C}_{\hat{\lambda}}$ has coverage rate at least $1 - \alpha - \tau$. Note that while $l$ does not explicitly appear in the guarantee, larger calibration sets lead to more precise $p$-value estimates, typically resulting in a larger $\Lambda_{\text{val}}$ and less conservative interval selection. In practice, any pair $(a, b) \in \Lambda_{\text{val}}$ can be used to construct prediction intervals. Although the quantile levels $c, d$ from Definition 3 could also be tuned jointly with $(a, b)$, we fix $c, d$ for simplicity, which is typically sufficient in our experiments.

We note that $\Lambda_{\text{val}}$ can be empty, producing no interval. This is intentional, as it can indicate that the shift is too large and retraining is needed, rather than producing an unreliable interval. In this setting, fixed $c, d$ offer a clear interpretation as stage-wise sensitivity to shifts (Section 7.1). While the coverage guarantees assume IID calibration, the selective FWER process creates a robustness buffer, which we extend to targeted adaptation of the selected weights when distribution shifts affect pipeline stages asymmetrically, which we define in Section 6 and demonstrate experimentally in Section 7. Lastly, Appendix C.2 shows how the IID assumption on $S^{\text{cal}}$ can be relaxed to allow stationary $\phi$-mixing sequences as an initial step to extending the theoretical coverage guarantees to the distribution shift settings we consider in Section 7.

## 6 ADAPTIVE RISK CONTROL WITH RESIDUAL DECOMPOSITION

We consider an adaptive variant of our method for nonstationary data by updating the prediction intervals over time. The sets $S^{\text{conf}}$ and $S^{\text{cal}}$ are defined using a sliding window over the most recent observations, with a user-specified window-length $k$. We construct intervals combining Definition 4 and Definition 3: $\hat{C}_{\alpha,a,b,c,d}(w) = \hat{\mu}_2(\hat{\mu}_1(w)) \pm (a\,Q_{1-c}(\{\Delta R_1\}) + b\,Q_{1-d}(\{R_2\}))$, parameterized by scaling coefficients $a, b$, quantile levels $c, d$, and target coverage level $\alpha$, where quantiles are taken over $S^{\text{conf}}$, and $\Lambda_{\text{val}}^t$ is recalculated at each time step. With each new point, we dynamically update the $(a_t, b_t, c_t, d_t, \alpha_t)$ using adaptive rules based on recent performance: when coverage drops below target, we decrease $\alpha_t$ and vice versa; when specific components show persistent errors we adjust their corresponding scaling parameters; and when $\Lambda_{\text{val}}^t$ is too restrictive we adjust the quantile levels to expand future options. The algorithmic details are provided in Appendix Section B.5.

Crucially, the SELECTLAMBDA algorithm implements the adaptive adjustments to $(a_t, b_t, c_t, d_t)$: it first identifies which component constitutes more of the error by comparing recent averages $\bar{\Delta R}_1 = \frac{1}{k}\sum_{i=t-k}^{t-1} \Delta R_1^{(i)}$ and $\bar{R}_2 = \frac{1}{k}\sum_{i=t-k}^{t-1} R_2^{(i)}$ over the sliding window. If coverage fails and upstream errors dominate ($\bar{\Delta R}_1 > \bar{R}_2$), it seeks to increase $a_t$ within the validated set $\Lambda_{\text{val}}^t$. Conversely, when downstream errors dominate ($\bar{R}_2 > \bar{\Delta R}_1$), it prioritizes $b_t$. If the desired scaling adjustment is unavailable in $\Lambda_{\text{val}}^t$, the algorithm returns signals $\Delta c_t, \Delta d_t \in \{-1, 0, 1\}$ to adjust $c_{t+1}, d_{t+1}$, effectively "adjusting" the constraints to allow more suitable scaling options.

Even with additional parameters, this approach preserves the long-run coverage guarantee from prior work: $\lim_{T\to\infty} \frac{1}{T}\sum_{t=1}^{T} \text{cov}_t \xrightarrow{\text{a.s.}} 1 - \alpha$, where $\text{cov}_t$ denotes coverage at time $t$. This is because the core $\alpha_t$ update rule remains identical to the adaptive conformal method of Gibbs and Candès (2021), resulting in the convergence guarantee formally stated in Appendix A.1.1.

## 7 EXPERIMENTS

We evaluate our method on synthetic and real-world forecasting tasks to assess its ability to (i) preserve coverage, (ii) attribute predictive error to specific model stages, and (iii) adapt to distribution shifts in modular pipelines. We validate the shift robustness mechanisms outlined previously: conservative calibration via FWER provides a safety margin, while stage-wise parameters $(a, b, c, d)$ adapt to distribution shifts that affect upstream and downstream components differently. Our experiments are designed to isolate upstream, downstream, and full-pipeline shifts, highlighting how stage-aware intervals improve transparency and robustness over existing conformal baselines. Additional experiments and details are in Appendix D where we provide ablation studies over hyperparameters, practical guidelines on choosing hyperparameter values, experiments on covariate shift, experiments on a real-world stocks dataset, and further visualizations, particularly how the chosen weights $(a, b)$ change due to distribution shifts.

### 7.1 NON-ADAPTIVE METHODS

We evaluate our non-adaptive method (Section 4), using the FWER-based procedure for our unified interval which we denote as $\text{SR}_{a,b}$ to generate a validated set $\Lambda_{\text{val}}$ of $(a, b)$, using fixed quantile levels $c, d$. We then select from this set the pair $(a, b)$ that yields coverage closest to the nominal level $\alpha = 0.1$ on $S^{\text{conf}}$. We compare against two nonadaptive baselines: standard split conformal prediction

Table 1: Average coverage and interval width for upstream and downstream shifts under gradual and rapid shifts at $\alpha = 0.1$, over 50 samples, with std for SR, SC, WSC methods. NA indicates that our method abstains from providing an interval

(a) Upstream shift coverage and width

| Shift | Metric | $SR_{a,b}$ (c=d=0.01) | $SR_{a,b}$ (c=0.05,d=0.01) | SC | WSC |
|---|---|---|---|---|---|
| Gradual | Coverage | $0.8419 \pm 0.01$ | $0.8233 \pm 0.01$ | $0.7280 \pm 0.02$ | $0.7488 \pm 0.02$ |
| | Width | $2.9884 \pm 0.10$ | $2.4272 \pm 0.10$ | $2.3726 \pm 0.10$ | $2.4849 \pm 0.09$ |
| Rapid | Coverage | $0.8041 \pm 0.01$ | NA | $0.6910 \pm 0.01$ | $0.7218 \pm 0.01$ |
| | Width | $3.9703 \pm 0.17$ | NA | $2.9544 \pm 0.11$ | $3.1969 \pm 0.15$ |

(b) Downstream shift coverage and width

| Shift | Metric | $SR_{a,b}$ (c=d=0.01) | $SR_{a,b}$ (c=0.01,d=0.05) | SC | WSC |
|---|---|---|---|---|---|
| Gradual | Coverage | $0.9309 \pm 0.02$ | $0.9133 \pm 0.02$ | $0.8695 \pm 0.03$ | $0.8748 \pm 0.02$ |
| | Width | $2.4274 \pm 0.12$ | $2.3738 \pm 0.12$ | $2.0143 \pm 0.06$ | $2.0472 \pm 0.06$ |
| Rapid | Coverage | $0.8981 \pm 0.01$ | NA | $0.8419 \pm 0.02$ | $0.8506 \pm 0.02$ |
| | Width | $2.4878 \pm 0.11$ | NA | $2.1035 \pm 0.08$ | $2.1498 \pm 0.08$ |

(SC, Vovk et al. (2005)) and weighted split conformal prediction (WSC, Barber et al. (2023)), which accounts for non-exchangeability.

We begin with a synthetic dataset generated by the structural equations $x = 3w + \nu_1$, $y = 4x + \nu_2$, where $w, \nu_1, \nu_2 \sim \mathcal{N}(0, 0.1)$. After an initial stationary period, the data undergoes distribution shift in either the upstream or downstream with increasing noise. We vary the rate of these shifts to include both gradual and rapid changes. Results for both upstream and downstream shifts are shown in Table 1. Our method achieves higher coverage than the other methods, which suffer degradation under distributional shifts, particularly under upstream. For example, our method drops from 0.84 to 0.80 coverage vs. SC dropping from 0.73 to 0.69 for gradual vs. rapid upstream shifts. These coverage improvements are at the cost of wider intervals and occasional abstention; this trade-off is worthwhile when attribution and robustness are prioritized. We note that selecting larger $c$ (resp. $d$) increases the sensitivity of the method to upstream (resp. downstream shift), resulting in abstention under rapid shifts. This is intentional, with abstention signaling that retraining may be necessary for the given stage; black-box methods lack this ability, as they cannot isolate which part of the model fails, resulting in unnecessary retraining.

Thus, $c$ and $d$ control sensitivity: by selecting larger or smaller $c$ and $d$, practitioners can balance robustness and abstention at each stage, offering flexibility beyond traditional conformal approaches. Next, we evaluate the adaptive version of our method under more extreme distributional shifts.

### 7.2 Adaptive methods under structured distribution shifts

We evaluate our adaptive method (Section 6) in settings with drastic distribution shifts, comparing against recent adaptive baselines: Adaptive Conformal Inference (ACI, Gibbs and Candès (2021)), Proportional-Integral-Derivative control (PID, Angelopoulos et al. (2023)), and Online Conformal Inference with Decaying step sizes (OCID, Angelopoulos et al. (2024)). We also consider DtACI (Gibbs and Candès (2024)), an extension of ACI, and report its results in Appendix D.4.2 due to weaker performance. While DtACI is designed for online adaptation, it may be less effective in fixed-horizon forecasting tasks where the lag between prediction and feedback complicates adaptation. We also consider covariate shifts, hyperparameter sweeps, and a real-world stock price dataset, with full experimental details in Appendix D.

**Controlled two-stage regression with targeted shifts.** We simulate a two-stage regression pipeline, with controlled stochastic relationships between upstream input $w$, intermediate representation $x$, and downstream output $y$. The initial data follows an i.i.d. structure with $x = \mu_1(w) = 3w + \nu_1$, $y = \mu_2(x) = 4x + \nu_2$ where $\nu_1, \nu_2 \sim \mathcal{N}(0, 1)$. Both stages are modeled using ordinary least squares. At test time, we simulate a temporal sequence of three phases: (i) **Upstream distribution shift:** $\mu_1$ becomes $\mu_1^{\text{shift}}(w) = 8w + 1 + \nu_1$; (ii) **Reversion:** The upstream returns to $\mu_1(w)$; (iii) **Downstream**

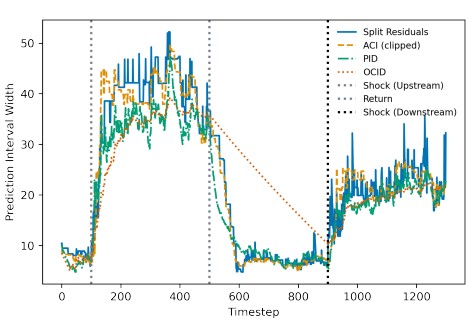 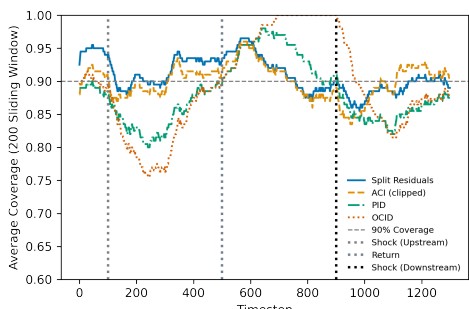

(a) Widths of the four intervals for $\alpha = 0.1$, $\delta = 0.1$, $\gamma = 0.01$, $\eta = 0.01$, $k = 100$

(b) Coverage averaged over sliding window of 200 points

Figure 2: Comparison of interval width and coverage under synthetic distribution shifts

**distribution shift:** $\mu_2$ becomes $\mu_2^{\text{shift}}(x) = 7x + 5 + \nu_2$. We report coverage and widths in Figure 2, capturing the robustness of our methods to stage-specific distribution shifts without overcompensating on width, particularly for upstream shifts.

**Automobile indicators dataset.** We evaluate our method on a real-world dataset consisting of monthly economic indicators for the U.S. automobile supply chain and used vehicle valuations. Upstream features $(w)$ are sourced from FRED (2025), including metrics such as price indices, inventory levels, and manufacturing prices, which are then forecasted $(x)$ at a 6-month horizon. Downstream outputs $(y)$, aggregated used car prices, are obtained from Manheim (2025). This setting fits the two-stage modeling framework: upstream economic conditions influence downstream pricing, and training a separate upstream predictor leverages more widely available historical supply chain data.

A notable feature of this dataset is the sharp distribution shift in 2020 due to the COVID-19 pandemic, which disrupted global supply chains, leading to prolonged production halts, resulting in a steep and persistent rise in used vehicle prices (Mullin, 2022). The nature of the distribution shift may also affect the model asymmetrically, leading to exacerbated upstream inaccuracy, which we observe via the scaling parameter changes in Appendix D.4.5. For the upstream forecasting task, we use an ensemble of VAR and ARIMA models to predict supply-chain indicators. The model is trained on a rolling-window basis, with residual scores computed at each time step. A 40-month sliding window forms the held-out set from which we derive $S^{\text{conf}}$ and $S^{\text{cal}}$.

We present the resulting prediction intervals for our method, ACI, PID, and OCID in Figure 3. Our method effectively responds to the distribution shift in 2020, maintaining coverage relative to other methods. OCID is also able to capture the distribution shift, but shows inefficiency during 2014-2019. Notably, our approach maintains a comparable average width (21.085 vs 15.696 (ACI), 18.469 (PID), and 31.058 (OCID)) and can quickly adjust to distribution shifts by consistently considering potential conservative values of $\lambda$ in $\Lambda_{\text{val}}$. This suggests that our method is well-suited for real-world forecasting and shines under asymmetric structural distribution shifts. In contrast, for covariate shifts (Appendix D) that affect the entire pipeline, the benefits are less significant.

## 8 CONCLUSION

We proposed a conformal prediction framework for sequential models that decomposes prediction residuals into distinct stage-specific components. Our method combines this decomposition with risk-controlled parameter selection to construct prediction intervals that provide both coverage guarantees and uncertainty attribution. The approach allows practitioners to identify which pipeline stage contributes to prediction uncertainty and provides diagnostic tools for targeted model improvement. Empirical evaluation on synthetic distribution shifts and real-world economic forecasting shows that our method's coverage degrades less severely under shifts compared to standard conformal approaches. While this robustness comes at the cost of wider intervals and occasional abstention when shifts are severe, these trade-offs reflect the method's conservative design that prioritizes reliable

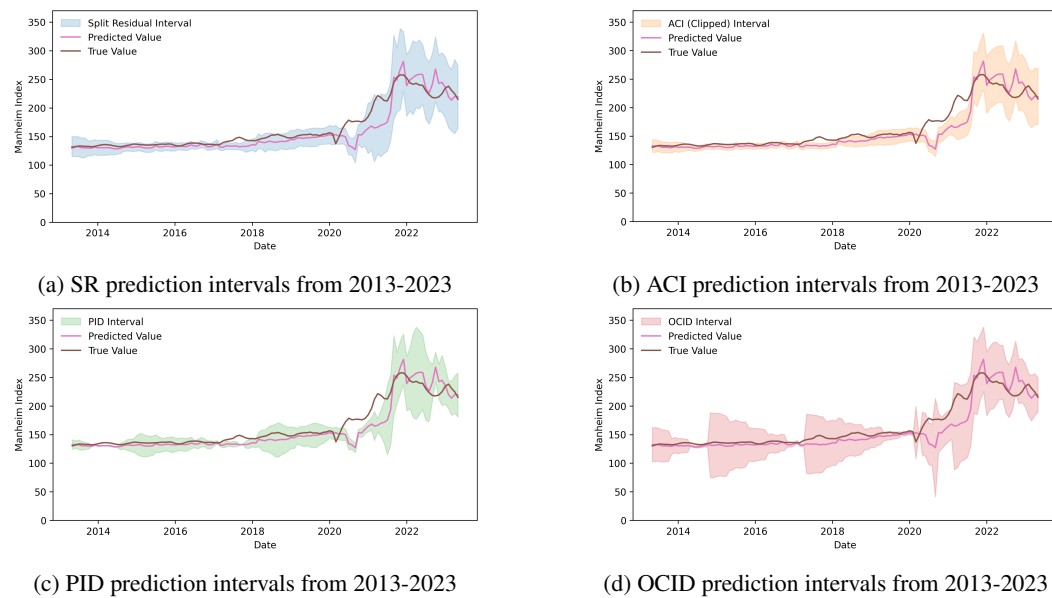

(a) SR prediction intervals from 2013-2023

(b) ACI prediction intervals from 2013-2023

(c) PID prediction intervals from 2013-2023

(d) OCID prediction intervals from 2013-2023

Figure 3: Coverage of prediction intervals for $\alpha = 0.2$, $\delta = 0.3$, $\gamma = 0.01$, $\eta = 0.01$, $k = 40$

coverage over optimistic predictions. The stage-wise decomposition proves particularly valuable for asymmetric shifts affecting different pipeline components, enabling targeted adaptive responses that standard methods cannot provide.

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

## A APPENDIX

### A.1 MAIN RESULTS/PROOFS

**Proof of Theorem 1**

*Proof.* We consider the event

$$y_{test} \notin \hat{C}_{c,d}(w_{test})$$

where $\hat{C}_{c,d}$ is the prediction interval defined by $\hat{\mu}_2(\hat{\mu}_1(w_{test})) \pm Q_{1-c}(\{\Delta R_1\}) + Q_{1-d}(\{R_2\})$ with the quantiles being taken over the residual components calculated on held-out conformal set $S^{\text{conf}}$. This event is when the coverage of the interval fails. See that

$$y_{test} \notin \hat{C}_{c,d} \implies |y_{test} - \hat{\mu}_2(\hat{\mu}_1(w_{test}))| \geq Q_{1-c}(\{\Delta R_1\}) + Q_{1-d}(\{R_2\})$$

But then by definition

$$|y_{test} - \hat{\mu}_2(\hat{\mu}_1(w_{test}))| \geq Q_{1-c}(\{\Delta R_1\}) + Q_{1-d}(\{R_2\}) \implies$$

$$||y_{test} - \hat{\mu}_2(x_{test})| - |y_{test} - \hat{\mu}_2(\hat{\mu}_1(w_{test}))|| + |y_{test} - \hat{\mu}_2(x_{test})| \geq Q_{1-c}(\{\Delta R_1\}) + Q_{1-d}(\{R_2\})$$

Thus, the split residual on the *test* point also falls outside the region. However, this implies that at least one of the following events occur:

$$A_1 : |y_{test} - \hat{\mu}_2(x_{test})| \geq Q_{1-d}(\{R_2\})$$

or

$$A_2 : ||y_{test} - \hat{\mu}_2(x_{test})| - |y_{test} - \hat{\mu}_2(\hat{\mu}_1(w_{test}))|| \geq Q_{1-c}(\{\Delta R_1\})$$

Then each of these pieces hold due to exchangeable properties as the probability of the new residual piece being higher in rank than the $1-c$ quantile (resp. $1-d$) is $c$ (resp. $d$). For $R_2$, the exchangeability is clear as it is directly the prediction interval for $\mu_2$ on data $(x, y)$. For the latter, it is still clearly exchangeable as well: the functions $\hat{\mu}_1, \hat{\mu}_2$ are symmetric and deterministic and are pretrained, thus being fixed on the held-out set. Thus $\Delta R_1$ can be interpreted as the result of some deterministic function $\psi$ on each point $(w_i, x_i, y_i)$, which implies that

$$P(\Delta R_1^1 = r_1, \ldots, \Delta R_1^m = r_m)$$
$$= P((w_1, x_1, y_1) \in \psi^{-1}(r_1), \ldots, (w_m, x_m, y_m) \in \psi^{-1}(r_m))$$
$$= P((w_{\pi(1)}, x_{\pi(1)}, y_{\pi(1)}) \in \psi^{-1}(r_{\pi(1)}), \ldots, (w_{\pi(m)}, x_{\pi(m)}, y_{\pi(m)}) \in \psi^{-1}(r_{\pi(m)}))$$
$$= P(\Delta R_1^{\pi(1)} = r_{\pi(1)}, \ldots, \Delta R_1^{\pi(m)} = r_{\pi(m)})$$

where $\Delta R_1^i$ denotes the first residual component of the $i$-th point of $S^{\text{conf}}$. Then we have exchangeability of $\Delta R_1$, thus the probability of each of the noncontainment events for each residual component are bounded by $c$ and $d$ respectively.

Therefore

$$P(y_{test} \notin \hat{C}_{c,d}) \leq c + d - P(A_1 \cap A_2) \leq c + d.$$

In fact, as can be seen in the above equation, the stated result is looser than necessary. □

**Proof of Proposition 2**

*Proof.* Since the traditional residual $R$ is assumed to be continuous, the function that maps the scaling parameters $(a, b)$ to the marginal coverage probability of the interval $\hat{C}_{\alpha/2,a,b}$ is itself continuous. When $a = b = 0$, the interval degenerates to a single point and thus has zero coverage (assuming the problem is nontrivial). When $a = b = 1$, the interval is wide enough to ensure coverage of at least $1 - \alpha$ by Corollary 1. By the intermediate value theorem, there must exist a pair $(a^*, b^*) \in [0, 1]^2$ such that the interval achieves exactly $1 - \alpha$ coverage. As an aside, one can further view the problem as an optimization problem

$$\min_{a,b} aQ_{1-\alpha}(\{\Delta R_1\}) + bQ_{1-\alpha}(\{R_2\})$$
$$s.t. \quad P(R \geq aQ_{1-\alpha}(\{\Delta R_1\}) + bQ_{1-\alpha}(\{R_2\})) \leq \alpha$$

which by KKT conditions implies that any $a^*, b^*$ must satisfy

$$\frac{1}{Q_{1-\alpha}(\{\Delta R_1\})} \frac{\partial P(R \geq aQ_{1-\alpha}(\{\Delta R_1\}) + bQ_{1-\alpha}(\{R_2\})))}{\partial a}$$
$$= \frac{1}{Q_{1-\alpha}(\{R_2\})} \frac{\partial P(R \geq aQ_{1-\alpha}(\{\Delta R_1\}) + bQ_{1-\alpha}(\{R_2\})))}{\partial b}$$

which provides that at any optimal solution $a^*, b^*$ there must be the same balance between the magnitude of components and the impact (derivative) of the scaling parameter on true coverage probability for each parameter. Thus large magnitude implies large impact, confirming intuitive understanding. □

**Proof of Theorem 2**

*Proof.* By Lemma A.2, the $p$-values are super-uniform under the null hypothesis. Then we may apply Theorem 1 of Angelopoulos et al. (2021) (Theorem 3), given a FWER algorithm $\mathcal{A}$ with parameter $\delta > 0$ to obtain the result. We note that the size of $S^{\text{cal}}$ does not explicitly appear in the bound, however, it directly affects the $p$-values which can be seen via Hoeffding's inequality:

$$P(\alpha - \text{Bin}(l, \alpha) \geq \alpha - l\hat{\mathcal{R}}(\lambda)) \leq \exp\left(\frac{-2(\alpha - l\hat{\mathcal{R}}(\lambda))^2}{l}\right)$$

where $\hat{\mathcal{R}}$ is the empirical error rate on $S^{\text{cal}}$ □

### A.1.1 Long-Run Coverage

**Proposition 3** (Long-Run Coverage of Adaptive Method). *Let $\gamma > 0$ be the step size, $\alpha \in (0,1)$ the nominal coverage level, and let $cov_t \in \{0,1\}$ denote the coverage indicator at time $t$. Then the adaptive algorithm satisfies:*

$$\lim_{T \to \infty} \frac{1}{T} \sum_{t=1}^{T} cov_t \xrightarrow{a.s.} 1 - \alpha.$$

*This result ensures that in the long run, the algorithm achieves the desired coverage level $1 - \alpha$, without requiring distributional assumptions on the data.*

The additional parameters $(a_t, b_t, c_t, d_t)$ are updated as deterministic functions of the algorithm's history, ensuring they do not interfere with the update steps that underlie the coverage guarantee.

A finite-sample convergence bound (Appendix Proposition 6) extends existing Markov chain analysis to include our additional parameters, with the key insight that the monotonicity properties required still hold under our parameter update rules.

**Proof of Proposition 3** Please refer to Algorithm 1 for notation. We first show a preliminary lemma.

**Lemma A.1** (Boundedness of $c_t$ and $d_t$). *Let $\eta > 0$ be a constant. Then, for all $t \in \mathbb{N}$, the sequences $c_t$ and $d_t$ are bounded within the interval $[-\eta, 1+\eta]$. That is,*

$$\forall t \in \mathbb{N}, \quad -\eta \le c_t \le 1+\eta \quad and \quad -\eta \le d_t \le 1+\eta.$$

*Proof.* To establish the lower bound for $c_t$, observe that the argument for $d_t$ follows similarly, and the upper-bound can be treated identically.

Suppose that $c_t < 0$ for some time step $t$. In this case, by the definition of the residual component $\Delta R_1$, we have:

$$\Delta R_1 = \infty \implies cov_t = 1,$$

where $cov_t$ denotes the coverage of the model at time $t$. Additionally, since $cov_t^{\Delta R_1} = 0$ by the construction of the residual, this implies that in the output of the algorithm SELECTLAMBDA, the update step $\Delta c_t \ge 0$, meaning that:

$$c_{t+1} \ge c_t.$$

Thus, if $c_t$ were negative at some time step, the next value is non-decreasing. Consequently, the sequence $c_t$ cannot decrease below by $-\eta$. Therefore, the minimum value attainable by $c_t$ is $-\eta$.

A similar argument holds for the upper-bound.

Hence, we conclude that:

$$-\eta \le c_t \le 1+\eta \quad and \quad -\eta \le d_t \le 1+\eta \quad \forall t \in \mathbb{N}.$$

This holds similarly for $d_t$. $\square$

Next, we show the asymptotic result, Section A.1.1 via the same arguments as Proposition 4.1 of Gibbs and Candès (2021).

*Proof.* Although our algorithm includes additional parameters $(a_t, b_t, c_t, d_t)$, the update rule for $\alpha_t$ remains unchanged from Gibbs and Candès (2021):

$$\alpha_t = \alpha_{t-1} - \gamma((1 - cov_{t-1}) - \alpha),$$

where $cov_t$ is the coverage indicator at time $t$ and $\alpha$ is the target coverage level.

As in Gibbs and Candès (2021), we show that $\alpha_t \in [-\gamma, 1+\gamma]$ with probability 1. For the lower bound (the upper-bound follows symmetrically), suppose $\alpha_t < 0$ at some time $t$. Then the candidate set $\Lambda_{\text{val}} = \emptyset$, so $cov_t = 1$, and the update becomes:

$$\alpha_{t+1} = \alpha_t + \gamma\alpha > \alpha_t.$$

Hence, $\alpha_t$ is pushed upward, preventing it from decreasing without bound. Therefore, $\alpha_t \geq -\gamma$ for all $t$.

Next, unfolding the recursion gives:

$$\alpha_T = \alpha_1 - \sum_{t=1}^{T-1} \gamma((1 - \text{cov}_t) - \alpha).$$

Rearranging terms and using the bound, we obtain:

$$\left| \frac{1}{T} \sum_{t=1}^{T} (1 - \text{cov}_t) - \alpha \right| \leq \frac{\max(\alpha_1, 1 - \alpha_1) + \gamma}{\gamma T}.$$

Equivalently:

$$\left| \frac{1}{T} \sum_{t=1}^{T} \text{cov}_t - (1 - \alpha) \right| \leq \frac{\max(\alpha_1, 1 - \alpha_1) + \gamma}{\gamma T},$$

which converges to zero as $T \to \infty$, yielding the desired result. $\square$

### A.2 AUXILIARY RESULTS

**Proposition 4.** *For a given point $(w, x, y)$, if $R_2 \leq \epsilon < R$ for some small $\epsilon > 0$ i.e. $\mu_2(x)$ is close to the true value $y$, then $|\Delta R_1 - R| \leq \epsilon$*

*Proof.* This follows immediately by definition of $\Delta R_1$.

$$|\Delta R_1 - R| = ||R_2 - R| - R|$$
$$\leq \epsilon$$

$\square$

Next, we show the equivalent for $R_2$ under additional assumptions

**Proposition 5.** *For a given point $(w, x, y)$, $|R_2 - R| < \epsilon$ for small $\epsilon > 0$ under certain assumptions, such as if $\hat{\mu}_2$ is a Lipschitz continuous function for some parameter $L$, and the first stage has small prediction error, $|x - \hat{\mu}_1(w)| < \delta$. Then $|R_2 - R| \leq L\delta$*

*Proof.*

$$|R_2 - R| \leq |\hat{\mu}_2(x) - \hat{\mu}_2(\hat{\mu}_1(w))|$$
$$\leq L|x - \hat{\mu}_1(w)|$$
$$\leq L\delta$$

$\square$

The main takeaway of these two results is that they confirm the intuition for when $\Delta R_1$ and $R_2$ are roughly similar to $R$, particularly when one model is accurate (and in the case of $R_2$ when the learned function is smooth). This also implies that both $\Delta R_1, R_2$ being small is impossible, thus one must represent a majority of the error. Note that $\Delta R_1$ and $R_2$ can also serve as rough lower bounds on $R$, though each can individually exceed $R$ under adversarial noise or model miscalibration.

**Lemma A.2.** *The $p_\lambda$ defined as $p_\lambda = \mathbb{P}\left( Bin(n, \alpha) \leq n\hat{\mathcal{R}}(\lambda) \right)$ are super-uniform.*

*Proof.* See Appendix D.1 of Quach et al. (2023). $\square$

**Theorem 3** (Theorem 1 of Angelopoulos et al. (2021))**.** *Suppose $p_\lambda$ has a distribution stochastically dominating the uniform distribution. Let $\mathcal{A}$ be a FWER controlling algorithm at level $\delta > 0$. Let $\mathcal{R}(\lambda)$ be the expected risk for the choice of $\lambda$ and $\alpha > 0$ be the maximal error rate. Then $\Lambda_{val} = \mathcal{A}(\{p_\lambda\}_{\lambda \in \Lambda})$ satisfies the following*

$$\mathbb{P}\left( \sup_{\lambda \in \Lambda_{val}} \{\mathcal{R}(\lambda)\} \geq \alpha \right) \leq \delta$$

## A.3 EXTENSIONS OF THEOREM 1

We can also state Theorem 1 for the setting with auxiliary data (Section C.3).

**Corollary 2.** *For $c, d \in (0, 1)$ and new point $(w_{test}, x_{test}, x'_{test}, y_{test})$, and $\hat{C}_{c,d}$ interval defined as in Section C.3 to include auxiliary data, we have*

$$P(y_{test} \in \hat{C}_{c,d}(w_{test})) \geq 1 - c - d.$$

which follows by the same proof as above; auxiliary data does not change the proof (assuming that with auxiliary features, the data is still exchangeable). Lastly, we note that the prior results can easily be adapted to nonexchangable data settings using weighted quantiles and a weighting scheme as mentioned in Barber et al. (2023). Thus we have

**Corollary 3.** *Let $|S^{conf}| = m$ and $p_1 \ldots, p_m$ be weights for each point. For $c, d \in (0, 1)$ and new point $(w_{test}, x_{test}, y_{test})$, and $\hat{C}_{c,d}$ interval defined as in Definition 3, we have*

$$P(y_{test} \in \hat{C}_{c,d}) \geq 1 - c - d - \psi_1 - \psi_2.$$

*where $\psi_1$ and $\psi_2$ are coverage penalties from nonexchangability.*

*Proof.* The result follows analogously to the proof of Section A.1, with modifications to account for the use of weighted quantiles and associated penalty terms. Specifically, we let $\psi_1$ and $\psi_2$ denote penalties applied to the empirical weighted quantiles, as introduced in Theorem 2 of Barber et al. (2023). These penalties are of the form $\sum_{i=1}^{m} \tilde{p}_i d_{TV}(\vec{R}, \vec{R}^i))$ where $\tilde{p}_i = \frac{p_i}{1 + \sum_{i=1}^{m} p_i}$, $\vec{R}$ represents the vector of full residuals and $\vec{R}^i$ is the same vector with the $i$-th point swapped with $(w_{test}, y_{test}, z_{test})$.

When applying weighted conformal prediction, the standard quantile threshold used in the unweighted case is replaced with a weighted quantile. By Theorem 2 of Barber et al. (2023), these penalties $\psi_1, \psi_2$ ensure that the resulting prediction set retains valid marginal coverage.

Therefore, by adapting the same steps as in Section A.1—but replacing the empirical quantiles with weighted quantiles and applying Theorem 2 of Barber et al. (2023), we obtain the stated result. $\square$

We also introduce a result for coverage for fixed scaling values of $a, b$ under some assumptions.

**Corollary 4.** *Assume that the CDF of the residual for an out-of-sample point is Lipschitz continuous with constant $L$. Furthermore, if the maximal value of $\{\Delta R_1\}$ (resp. $\{R_2\}$) is bounded by $\epsilon > 0$, then if we set $a = 0, b = 1$ (resp. $a = 1, b = 0$) with $c = d = \alpha$, we have coverage with*

$$P(y_{test} \notin \hat{C}_{0,1,\alpha}(w_{test})) \leq 2\alpha + L\epsilon.$$

*Proof.* Consider $\hat{C}_{1,1,\alpha}(w_{test}) = Q_{1-\alpha}(\{\Delta R_1\}) + Q_{1-\alpha}(\{R_2\})$, which has coverage guarantee

$$P(y_{test} \notin \hat{C}_{1,1,\alpha}(w_{test})) \leq 2\alpha.$$

By assumption, $Q_{1-\alpha}(\{\Delta R_1\}) < \epsilon$, and thus instead using $\hat{C}_{0,1,\alpha}(w_{test})$ results in a change of at most $\epsilon$, which by Lipschitz continuity of the CDF produces a resultant change of $L\epsilon$ in the probability guarantee. $\square$

The existence of the Lipschitz condition is quite reasonable, for example, we consider $\mu_2$ is a linear function and the noise term is normally distributed, the residuals themselves are half-normally distributed and thus satisfy the above requirement. We pay a small theoretical guarantee when shrinking the interval, under the assumption $L\epsilon$ is small, which is not necessarily true (and $L$ is generally unknown). However, this result gives the intuition that if one of the residual pieces contributes little to the overall error, we can remove it for little coverage cost and serves as part of the motivation for the adaptive algorithm. We find that for certain settings with unbalanced residual components, even simple heuristics such as $\max(Q_{1-c}(\{\Delta R_1\}), Q_{1-d}(\{R_2\}))$ where $c = d = \alpha$ achieves similar coverage to the full residuals.

We can obtain non-asymptotic bounds for the specific setting of a hidden Markov Model, as in Theorem 4.1 of Gibbs and Candès (2021) once again, with little changes due to the shared update step.

**Proposition 6.** *Suppose that the data is coming from a hidden Markov model, with underlying states $A_t \in \mathcal{A}$ such that the data $(w_t, x_t, y_t) \sim P_{A_t}$. Observe that $a_t, b_t, c_t, d_t, \alpha_t, A_t$ form a Markov chain on $[0,1]^2 \times [-\eta, 1+\eta]^2 \times [-\gamma, 1+\gamma] \times \mathcal{A}$ with stationary distribution $\pi$. Assume that the algorithm has passed its burn-in period and is now in a stationary setting. Suppose that the spectral gap of $\{A_t\} = 1 - \eta > 0$. Let $\mathrm{err}_t = 1 - cov_t$. Then let $B = \sup_{a \in \mathcal{A}} |E[\mathrm{err}_t | A_t = a] - \alpha|$ and $\sigma_B^2 = E[(E[\mathrm{err}_t | A_t] - \alpha)^2]$. Then*

$$P(|\frac{1}{T} \sum_{t=1}^T \mathrm{err}_t - \alpha| \geq \epsilon) \leq 2\exp(-T\epsilon^2/8) + 2\exp\left(\frac{-T(1-\eta)\epsilon^2}{8(1+\eta)\sigma_B^2 + 20B\epsilon}\right)$$

showing a finite-sample bound deviation bound for the empirical coverage.

*Proof.* The structure of this proof follows that of Theorem 4.1 in Gibbs and Candès (2021), which depends on their Lemma A.2 and Theorem A.1. We outline the necessary modifications to their lemma (written below) to hold in our setting.

**Lemma A.3.** *For any $\tau \in \mathbb{R}$ and $t \in \mathcal{N}$*

$$\mathbb{E}\left[\prod_{s=1}^t \exp(\tau(\mathrm{err}_s - \mathbb{E}[\mathrm{err}_s | A_s]))\right] \leq \exp(\tau^2/2)\mathbb{E}\left[\prod_{s=1}^{t-1} \exp(\tau(\mathrm{err}_s - \mathbb{E}[\mathrm{err}_s | A_s]))\right]$$

To show this, we condition on $(a_1, b_1, c_1, d_1)$ alongside $(\alpha_1, A_1, \ldots, A_t)$. As in Gibbs and Candès (2021), we require that

$$f(\sum_{s=1}^{t-1} \mathrm{err}_s) = \prod_{s=1}^{t-1} \exp(\tau(\mathrm{err}_s - \mathbb{E}[\mathrm{err}_s | A_s]))$$

is non-decreasing and

$$g(\sum_{s=1}^{t-1} \mathrm{err}_s) = \mathbb{E}\left[\exp(\tau(\mathrm{err}_t - \mathbb{E}[\mathrm{err}_t | A_t])) | \mathrm{err}_1, \ldots, \mathrm{err}_{t-1}\right] = \mathbb{P}_{A_t}(y_t \in \hat{C}_{a_t, b_t, c_t, d_t, \alpha_t}(w_t))\exp(-\tau\mathbb{E}[\mathrm{err}_t | A_t])$$

$$+ (1 - \mathbb{P}_{A_t}(y_t \in \hat{C}_{a_t, b_t, c_t, d_t, \alpha_t}(w_t)))\exp(\tau(1 - \mathbb{E}[\mathrm{err}_t | A_t]))$$

is non-increasing in $\sum_{s=1}^{t-1} \mathrm{err}_s$ for $\tau \geq 0$. This still holds because:

- $\alpha_t$ remains a non-increasing function of $\sum_{s=1}^{t-1} \mathrm{err}_s$ under the same update rule.

- $a, b_t, c_t, d_t$ are deterministic functions of the conditioning variables (including $\mathrm{err}_1, \ldots, \mathrm{err}_{t-1}$), as the candidate set selection and parameter updates are fixed given the history.

- As $\alpha_t$ decreases, the validation set $\Lambda_{\mathrm{val}}$ shrinks (selecting only higher-width pairs), so the interval width is non-decreasing, thus $\mathbb{P}_{A_t}(y_t \in \hat{C}_{a_t, b_t, c_t, d_t, \alpha_t}(w_t))$ is non-decreasing, meaning that $g$ is overall non-increasing.

These observations ensure that the monotonicity conditions in Lemma A.2 still hold under our conditioning. The remainder of the proof, including the application of the Markov chain concentration inequality (Theorem A.1 of Gibbs and Candès (2021)), proceeds unchanged. □

# B CONCEPTUAL INFORMATION

## B.1 DISCUSSION ON QUANTILES OF RESIDUAL COMPONENTS.

We observe that when quantiles of the residual components are taken then summed, they do not necessarily serve as upper-bounds on the corresponding quantiles of the full residuals, which we use in our implementation. Specifically, the inequality

$$Q_{1-\alpha}(\{\Delta R_1\}) + Q_{1-\alpha}(\{R_2\}) < Q_{1-\alpha}(\{R\})$$

may not hold. This observation is significant because while the triangle inequality holds for each individual point, i.e., $R \leq \Delta R_1 + R_2$, such a relationship does not extend to the quantiles. This discrepancy is intentional, as it allows for a tighter combination of residuals compared to the traditional residual. Moreover, in practice, this situation occurs rarely.

## B.2  NOTATION TABLE

We provide a table of notation in Table 2.

Table 2: Summary of Notation

| Symbol | Meaning |
| --- | --- |
| $\hat{\mu}_1$ | First stage hypothesis |
| $\hat{\mu}_2$ | Second stage hypothesis |
| $R$ | Residual $|y - \hat{\mu}_2(\hat{\mu}_1(w))|$ for two-stage model |
| $\Delta R_1$ | First stage residual component $| |y - \hat{\mu}_2(x)| - |y - \hat{\mu}_2(\hat{x})| |$ |
| $R_2$ | Second stage residual component $|y - \hat{\mu}_2(x)|$ |
| $\alpha$ | Desired Miscoverage Rate |
| $a$ | Scaled weight for $\Delta R_1$ |
| $b$ | Scaled weight for $R_2$ |
| $c$ | Quantile level to take of $\Delta R_1$ |
| $d$ | Quantile level to take of $R_2$ |
| $\delta$ | Rejection stringency to control FWER |
| $\gamma$ | Step-size for updating $\alpha_t$ with $\gamma = 0.01$ empirically performing well |
| $\eta$ | Step-size for updating $c_t, d_t$ with $\eta = 0.01$ performing well |
| $\tau$ | Tolerance for calculating $p$-values for hypothesis testing |
| $k$ | Window length for adaptive method, with stabilized performance above 100 points |

## B.3  CHOICE OF COMPONENTS

Certainly, one could instead take quantiles or rescalings of the total sum $\Delta R_1 + R_2$ instead of separating it into components, however we choose to separate it multiple reasons. First, we note that by splitting the quantiles/weights, it becomes clear in the decomposition which stage contributes more to the error. Second under co-monotonicity of $\Delta R_1, R_2$, the quantiles of $\Delta R_1, R_2$ match and thus both approaches result in the same interval $Q_{1-\alpha}(\{\Delta R_1 + R_2\})$ which satisfies coverage guarantees as it is an upper-bound on the black box width. Furthermore, for smaller miscoverage levels $\alpha_1, \alpha_2$ then $Q_{1-\alpha_1}(\{\Delta R_1\}) + Q_{1-\alpha_2}(\{R_2\})$ is an upperbound on $Q_{1-\alpha}(R)$, so our use of $Q_{1-\alpha}(\{\Delta R_1\}) + Q_{1-\alpha}(\{R_2\})$ can be an upper-bound. Lastly, we also desire the ability to be tighter than $Q_{1-\alpha}(R)$ at times, unlike $Q_{1-\alpha}(\{\Delta R_1 + R_2\})$, to be able to quickly adapt to easy settings.

## B.4  FWER-CONTROLLING ALGORITHMS FOR SCALING SELECTION

To construct the set of valid scaling parameter pairs $\Lambda_{\text{val}}$, we apply multiple hypothesis testing procedures that control the family-wise error rate (FWER). This ensures that, with high probability, none of the accepted scaling pairs exhibit a true miscoverage rate above the nominal level $\alpha$.

**Definition 5** (FWER-Controlling Algorithm). *Let $\Lambda = \{\lambda_1, \ldots, \lambda_u\}$ be a candidate set of scaling pairs, and let $p_1, \ldots, p_u \in [0, 1]$ denote the associated p-values testing the null hypothesis that the miscoverage of each $\lambda_i$ exceeds $\alpha$. A selection procedure $\mathcal{A} : [0, 1]^u \rightarrow 2^{\{1, \ldots, u\}}$ is said to control the family-wise error rate (FWER) at level $\delta$ if:*

$$\mathbb{P}\left(\mathcal{A}(p_1, \ldots, p_u) \subseteq J\right) \geq 1 - \delta,$$

*where $J \subseteq \{1, \ldots, u\}$ is the set of true null hypotheses.*

**Bonferroni Correction.**    A classical method for controlling FWER is the Bonferroni correction Bonferroni (1937). It accepts any $\lambda_i$ for which $p_i \leq \delta/u$, where $u = |\Lambda|$. By the union bound, this ensures that the probability of falsely including any $\lambda$ with miscoverage above $\alpha$ is at most $\delta$. However, Bonferroni can be overly conservative when the number of tests $u$ is large.

**Fixed Sequence Testing.** To improve power in large-scale settings, we adopt fixed sequence testing Bauer (1991). This procedure assumes a pre-specified ordering of hypotheses $p_{(1)}, p_{(2)}, \ldots, p_{(u)}$, based on prior knowledge or heuristics (e.g., larger values of $a + b$ are expected to have lower risk). Hypotheses are tested sequentially: each $p_{(i)}$ is compared to $\delta$, and the first failure (i.e., $p_{(i)} > \delta$) terminates the process. All previous hypotheses are accepted, and the rest are rejected. This method is more powerful than Bonferroni in the presence of a meaningful ordering.

**Practical Note.** In our algorithm, we use fixed sequence testing to select valid scaling parameters. If no candidate passes the test, i.e., $\Lambda_{\text{val}} = \emptyset$, we abstain from prediction. This outcome indicates a lack of statistical evidence and may signal that additional calibration data or model refinement is necessary.

### B.5 ADAPTIVE METHOD PSEUDOCODE

---

**Algorithm 1** Adaptive Risk Controlling with $\Delta R_1, R_2$

---

**Require:** Window size $k$, Step sizes $\gamma, \eta$; target coverage level $\alpha$
1: **for** time step $t$ **do**
2:     **Obtain** window of $k$ recent observations
3:     **Obtain** Parameters $\alpha_t, c_t, d_t$, previous coverage $\text{cov}_{t-1}, \text{cov}_{t-1}^{\Delta R_1}, \text{cov}_{t-1}^{R_2}, a_{t-1}, b_{t-1}$
4:     **Split** window into $S^{\text{conf}}$ and $S^{\text{cal}}$ and compute $\{\Delta R_1\}, \{R_2\}$
5:     **Compute** quantiles:
6:        $\Delta R_1 \leftarrow Q_{c_t}(\{\Delta R_1\}), \quad R_2 \leftarrow Q_{d_t}(\{R_2\})$
7:     **Determine** valid $\lambda$ set:
8:        $\Lambda_{\text{val}}^t \leftarrow \text{FIXEDSEQUENCETESTING}(S^{\text{cal}}, \alpha_t, \Delta R_1, R_2)$
9:     **Obtain** $a_t, b_t, \Delta c_t, \Delta d_t \leftarrow \text{SELECTLAMBDA}(\Lambda_{\text{val}}^t, \text{cov}_{t-1}, \text{cov}_{t-1}^{\Delta R_1}, \text{cov}_{t-1}^{R_2}, a_{t-1}, b_{t-1})$
10:    **Calculate** interval $\hat{C}_{\alpha, a_t, b_t, c_t, d_t}(w_t)$
11:    **Check** coverage at $t$: $\text{cov}_t, \text{cov}_t^{\Delta R_1}, \text{cov}_t^{R_2}$
12:    **Update**: $\alpha_{t+1} \leftarrow \alpha_t + \gamma(\alpha - \text{cov}_t)$
13:       $c_{t+1} \leftarrow c_t + \eta \Delta c_t, \quad d_{t+1} \leftarrow d_t + \eta \Delta d_t$
14: **end for**

---

We define the coverage indicator at time $t$ as $\text{cov}_t = \mathbb{1}_{y_t \in \hat{C}_{\alpha, a, b, c, d}(w_t)}$. To capture deviations in the score components, we define $\text{cov}_t^{\Delta R_1}$ to be 0 if the true value $(\Delta R_1)_t$ lies within the estimated quantile interval for $\Delta R_1$, and equal to the excess $(\Delta R_1)_t - \Delta R_1$ otherwise. This can be succinctly written as $\text{cov}_t^{\Delta R_1} = \text{ReLU}((\Delta R_1)_t - \Delta R_1)$, and similarly for $\text{cov}_t^{R_2}$.

## C EXTENSIONS

### C.1 ADDITIONAL HEURISTIC

One could alternatively define the residual components and combination as signed residual components to obtain tighter, asymmetric intervals, at the cost of coverage. We define new residual components

$$\tilde{R}_2 = y - \hat{\mu}_2(x)$$
$$\Delta \tilde{R}_1 = \hat{\mu}_2(\hat{\mu}_1(w)) - \hat{\mu}_2(x)$$

such that the sum of the components is exactly the full error rather than an absolute value upper-bound. Because the components are now signed, one should construct intervals differently, which also results in asymmetric intervals. Similarly to before, we can scale each component to produce intervals of the form

$$[\hat{\mu}_2(\hat{\mu}_1(w_{\text{test}}) + aQ_{c/2}(\{\Delta \tilde{R}_1\}) + bQ_{d/2}(\{\tilde{R}_2\}), \hat{\mu}_2(\hat{\mu}_1(w_{\text{test}}) + aQ_{1-c/2}(\{\Delta \tilde{R}_1\}) + bQ_{1-d/2}(\{\tilde{R}_2\})]$$

Experimentally in IID settings, it produces slightly tighter intervals without additional tolerance, however this heuristic is more prone to producing empty $\Lambda_{\text{val}}$ sets. For example, we report the

Table 3: Signed residual heuristic performance over varying $\alpha$

| Metric | $\alpha = 0.13$ | $\alpha = 0.10$ | $\alpha = 0.07$ | $\alpha = 0.05$ |
|---|---|---|---|---|
| Coverage | 0.9013±0.02 | 0.9196±0.01 | NA | NA |
| Width | 2.0265±0.13 | 2.1523±0.12 | NA | NA |

performance of the method under IID settings with varying levels of nominal error rate $\alpha$ in Table 3 exemplifying this behaviour. Thus we have chosen to use the heuristic given in the main body of the text, but we provide this as an option for many possible ways of constructing intervals using the residual decomposition we introduce. Furthermore, because the decomposition itself is flexible, one could incorporate any recent conformal prediction advancements into the ways quantiles are taken, such as weighted data, or in the adaptive case how the step-size and $\alpha_t$ etc. change.

### C.2 STATIONARY $\Phi$-MIXING

We describe how the FWER control method outlined in Section 5 can be extended beyond the IID setting to accommodate stationary $\phi$-mixing processes. Note that for dependent data, the concentration properties of empirical quantities degrade, resulting in looser tail bounds. This, in turn, inflates the values of the empirical error estimates $p_\lambda$, thereby requiring larger thresholds $\delta$ to ensure that the selected set $\Lambda_{\text{val}}$ remains nonempty.

Our goal is to identify values of $\lambda$ for which the expected coverage error is below a target level $\alpha$. For each $\lambda$, we consider the null hypothesis

$$H_0^\lambda : \quad \mathbb{P}\left(y_{\text{test}} \notin \hat{C}_\lambda(w_{\text{test}})\right) > \alpha,$$

and compute an empirical estimate of the coverage error using a held-out calibration dataset. To test this hypothesis, we apply a concentration inequality that bounds the deviation of the empirical error from its expectation under dependence. Specifically, we employ a result for bounded functions of $\phi$-mixing sequences from Kontorovich and Ramanan (2008), with refinements from Mohri and Rostamizadeh (2010), which provides suitable control over the error rates in the non-IID case.

**Theorem 4** (Mixing Concentration Inequality). *Let $Z_1, ..., Z_n$ be random variables distributed according to a $\phi$-mixing distribution. Let $f : Z^n \to R$ be a measurable function that is $c$-Lipschitz with respect to the Hamming metric for some $c > 0$. Then, for any $\epsilon > 0$, the following inequality holds:*

$$P(|f(Z_1, \ldots, Z_n) - \mathbb{E}[f(Z_1, \ldots, Z_n)]| \geq \epsilon) \leq 2 \exp\left(\frac{-2\epsilon^2}{nc^2 \|\Delta_n\|_\infty^2}\right)$$

*where* $\|\Delta_n\|_\infty = 1 + \sum_{i=1}^n \phi(i)$

We apply this theorem with $\hat{\mathcal{R}}(\lambda) = f((w_1, x_1, y_1), \ldots, (w_l, x_l, y_l)) = \frac{1}{l}\sum_{i=1}^l 1_{\{y_i \notin \hat{C}_\lambda(w_i)\}}$. Clearly this is $\frac{1}{l}$-Lipschitz with respect to Hamming metric. Furthermore,

$$\mathbb{E}[f(S^{\text{cal}})] = \frac{1}{l}\mathbb{E}[\sum_{i=1}^l 1_{\{y_i \notin \hat{C}_\lambda(w_i)\}}] = \frac{1}{l}\sum_{i=1}^l \mathbb{E}[1_{\{y_i \notin \hat{C}_\lambda(w_i)\}}] = \alpha$$

due to stationarity and the null hypothesis. Then

$$P(\mathbb{E}[\hat{\mathcal{R}}(\lambda)] - \hat{\mathcal{R}}(\lambda) \geq \epsilon) \leq 2 \exp\left(\frac{-2\epsilon^2}{l \|\Delta_l\|_\infty^2}\right)$$

where we substitute our realized $\alpha - \hat{\mathcal{R}}(\lambda)$ for $\epsilon$ and plug it into the RHS to obtain a $p$-value for $\lambda$.

Then we have the following:

**Corollary 5.** *Let $\Lambda_{val} \subseteq \Lambda$ be the set selected by a FWER-controlling algorithm at level $\delta$, based on p-values computed over $S^{cal}$ using the above method, which is assumed to be stationary $\phi$-mixing. Then, for any $\hat{\lambda} \in \Lambda_{val}$, we have*

$$\mathbb{P}\left(\mathbb{P}\left(y_{test} \notin \hat{C}_{\hat{\lambda}}(w_{test}) \,\middle|\, S^{cal}\right) \leq \alpha\right) \geq 1 - \delta,$$

*where the outer probability is over the randomness of $S^{cal}$, and the inner probability is over the test point $(w_{test}, x_{test}, y_{test})$ which is assumed to be drawn from the same stationary distribution.*

While the statement of the result is similar to before, note that $p_\lambda$ is an conservative upper-bound on the true $p$-value, thus the threshold to accept a $\lambda$ is more stringent. We show that the $p_\lambda$ are still super-uniform. Let $P$ be the random variable representing empirical risk when the error rate is $\alpha$ and $Q$ by the empirical risk when the error rate is $\alpha'$. Under the null hypothesis, $\alpha' > \alpha$, which implies that $Q$ stochastically dominates $P$ $(P(Q \leq t) \leq P(P \leq t))$. Let $f(q) = \exp\left(\frac{-2(\alpha-q)^2}{l\|\Delta_n\|_\infty^2}\right)$ and $F_P(t)$ and $F_Q(t)$ represent CDFs of $P$ and $Q$ respectively. Then

$$\begin{aligned}
p_\lambda &= f(Q) \\
&\geq P(\alpha - P \geq \alpha - Q) \\
&= F_P(Q)
\end{aligned}$$

Furthermore, letting $O = p_\lambda$, by the stochastic dominance assumption,

$$\begin{aligned}
P(O \leq o) &= P(p_\lambda \leq o) \\
&\leq P(F_P(Q) \leq o) \\
&\leq P(F_Q(Q) \leq o) \\
&= P(Q \leq F_Q^{-1}(o)) \\
&= F_Q(F_Q^{-1}(o)) \\
&= o
\end{aligned}$$

Then we apply a FWER-controlling algorithm such as one from Section B.4 to set the acceptance threshold to reject the null hypothesis such that the total type-1 error rate is less than $\delta$, then use Theorem 1 of Angelopoulos et al. (2021) (Theorem 3).

We note that it is possible to adapt this result to exchangeable sequences using specific concentration inequalities such as ones found in Serfling (1974) or more recently Barber (2024).

### C.3 DECOMPOSITION FOR AUXILIARY DATA

We can also define $\Delta R_1$ and $R_2$ when considering auxiliary data.

**Definition 6.** *If we have sample $S$ where data is of the form $(w, x, x', y)$ where $x'$ represents auxiliary data for the second stage such that we have learned upstream and downstream models $\hat{\mu}_1 : \mathcal{W} \to \mathcal{X}$, $\hat{\mu}_2 : \mathcal{X} \times \mathcal{X}' \to \mathcal{Y}$. Then we can decompose the residual into components as the following for some new point $(w, x, x', y)$*

$$\begin{aligned}
\Delta R_1(w, x, x', y) &= ||y - \hat{\mu}_2(x, x')| - |y - \hat{\mu}_2(\hat{\mu}_1(w), x')|| \\
R_2(w, x, x', y) &= |y - \hat{\mu}_2(x, x')|.
\end{aligned}$$

We can use these residual components to create prediction intervals analogously to the original setting without auxiliary data.

### C.4 DECOMPOSITION FOR MULTI-STAGE MODEL

Now we consider extending the definition of $\Delta R_1, R_2$ to more than two stages. We provide a straightforward extension, but a cleverer one may exist.

**Definition 7.** *Suppose we have sample $S$ where data is of the form $w_1, w_2, \ldots, w_{N+1}$ with $N$ stages and corresponding learned hypotheses $\hat{\mu}_i : \mathcal{W}_i \to \mathcal{W}_{i+1}$. Then for a point $(w_1, \ldots, w_{N+1})$, define*

$$\Delta R_1(w_1, \ldots, w_{N+1}) = |||w_{N+1} - \hat{\mu}_N(\hat{\mu}_{N-1}(\ldots \hat{\mu}_2(w_2)))| - |w_{N+1} - \hat{\mu}_N(\hat{\mu}_{N-1}(\ldots \hat{\mu}_1(w_1)))|||$$

$$\cdots$$

$$\Delta R_i(w_1, \ldots, w_{N+1}) = |||w_{N+1} - \hat{\mu}_N(\hat{\mu}_{N-1}(\ldots \hat{\mu}_{i+1}(w_{i+1})))| - |w_{N+1} - \hat{\mu}_N(\hat{\mu}_{N-1}(\ldots \hat{\mu}_i(w_i)))|||$$

$$\cdots$$

$$R_n(w_1, \ldots, w_{N+1}) = |w_{N+1} - \hat{\mu}_n(w_N)|$$

*Observe that the sum of these terms is still an upperbound on the "full" residual $R = |w_{N+1} - \hat{\mu}_N(\hat{\mu}_{N-1}(\ldots \hat{\mu}_1(w_1)))|$ by the triangle inequality and thus we have a residual decomposition.*

An issue with this definition is that if we try a hypothesis testing method and FWER algorithm to weight each of these components, the set $\Lambda$ becomes prohibitively large as it grows exponentially with each stage. A FWER algorithm that is able to intelligently search the possibilities will become necessary, whereas for two-stages, it was feasible to search across the entire grid. One way to mitigate this is to restrict the proposed set $\Lambda$ at each step by focusing only on the most important residual components, fixing the weight of the remaining ones to remain; i.e. select the top $u$ residual components with the highest average width in the window of recent observations and define $\Lambda$ only for those components, freezing the other component weights. In fact, one can sort the residual components by magnitude then sequentially search through $\Lambda$, freezing the others outside of the top $u$, resulting in only $uv$ tests, if $v$ is the number of sub-divisions for each component.

### C.5 DECOMPOSITION FOR MULTIPLE UPSTREAM MODELS

We consider an augmented two-stage model, with multiple upstream models each producing a component of $x$, the intermediate value, which is now vector valued. Suppose there are $N$ upstream models, $\hat{\mu}_1^1, \ldots, \hat{\mu}_N^1$ that each map upstream features $\mathbf{w} = w_1, \ldots, w_M$ to their corresponding intermediate feature in $\mathbf{x} = x_1, \ldots, x_N$. Denote the output of the $i$-th upstream model to be $\hat{x}_i$. Then we can define $\Delta R_1, R_2$ components for each downstream feature and take a weighted sum. Thus, for the $i$-th downstream feature, we have

$$(R_2)_i = |y - \hat{\mu}_2(\hat{x}_1, \ldots, x_i \ldots, \hat{x}_N)|$$
$$(\Delta R_1)_i = ||y - \hat{\mu}_2(\hat{x}_1, \ldots, \hat{x}_N)| - |y - \hat{\mu}_2(\hat{x}_1, \ldots, x_i \ldots, \hat{x}_N)||.$$

Then see that a convex combination $\sum_{i=1}^{nN} \theta_i((R_2)_i + (\Delta R_1)_i), \sum_{i=1}^{N} \theta_i = 1, \theta_i \in [0, 1]$ forms an upperbound on $R = |y - \hat{\mu}_2(\hat{x}_1, \ldots, \hat{x}_N)|$. This approach has a similar issue as the multi-stage model, as the search-space $\Lambda$ scales with the number of upstream models.

## D ADDITIONAL EXPERIMENTS

We provide additional experiments on the non-adaptive methods and adaptive methods, providing further insight into the parameters of the split residual method and the resulting coverage improvements. We include hyperparameter ablation studies as well as an additional real-world dataset. We include experiments on covariate shift to demonstrate that the split residual method is robust multiple types of shift and is not limited to stage-wise shift, which was extensively explored in the main text. All experiments were conducted on a laptop with an Intel i7 processor and 8GB of RAM. No specialized hardware (e.g., GPUs or TPUs) was used for training or evaluation.

### D.1 EXPERIMENTAL HYPERPARAMETER DETAILS

We describe the default hyperparameters we use for the other methods in our experiments: SC, WSC, ACI, DtACI, PID, OCID.

- SC: There is no other parameter used for split conformal prediction intervals other than the nominal desired miscoverage level $\alpha = 0.1$ which we keep the same across all methods
- WSC uses the same $\alpha = 0.1$ and also includes a weighting parameter 0.99 which exponentially weights points as they go further back in time, resulting in recent points having more weight.

- ACI uses the same initial $\alpha = 0.1$ and updates $\alpha_t$ using the same step size $\gamma = 0.01$ as well as the same window size $k = 100$ as our method.

- DtACI uses the same $\alpha = 0.1$ and also uses default multiple candidate values for step-size $\gamma = [0.001, 0.002, 0.004, 0.008, 0.0160, 0.032, 0.064, 0.128]$.

- PID uses the same initial $\alpha = 0.1$, step size $\gamma = 0.01$, and window size $k = 100$. Two additional parameters KI and $C_{\text{sat}}$ are estimated with appropriate hypothesized bound $B$ depending on the data, using the default heuristic given in Appendix B of Angelopoulos et al. (2023).

- OCID uses the same $\alpha = 0.1$ and decay parameter $0.1$ with the decay weight function given by Angelopoulos et al. (2024)

Furthermore, note that $S^{\text{cal}}$ for our method is a subset of the held-out data which the other methods have full access to, so we do not use additional data relative to the other methods but simply partition it into smaller sets.

## D.2 Hyperparameter Guidelines

We provide some simple guidelines for parameter settings for both the non-adaptive and adaptive case.

- A default $\Lambda$ grid: $\{(a, b) : a, b \in \{0.1, 0.2, 0.3, \ldots 1 - \alpha/2\}, a + b \geq \frac{1-\alpha}{2}\}$, which can be improved in granularity based on the size of calibration set $l$. A suggested granularity could be $l/20$ number of grid points. Experimentally, we used a default $\{0.1, 0.2, \ldots, 1\}$.

- Initial values of $c, d$ (or fixed values for the non-adaptive case) we recommend balanced $c = d \leq \alpha/2$. If one wishes to be more upstream-sensitive, $c = \alpha, d \leq \alpha/2$ and if one wishes to be downstream-sensitive, $c \leq \alpha/2, d = \alpha$.

- For $\delta$, a default choice of $0.3$ is for moderate conservativeness, but $\delta = 0.1$ is a more conservative choice that induces more abstention.

- For $\tau$ to be more conservative, we default to $\tau = 0$, but $\tau \in [0.03, 0.05]$ improves efficiency Table 5, so in "easier" regimes such as IID one should choose such values.

- For window-size $k$ for the adaptive method, we recommend $k > 40$, however, the more held-out data the more efficient values the method can consider.

- For step-size $\gamma$ we recommend a default of $0.01$ which has sufficed under a variety of scenarios, however a grid of $[0.01, 0.03, 0.05, 0.1]$ should cover all possible values.

- For step-size $\eta$ we recommend a default of $0.01$ for stable performance; higher values encourage more extreme behavior (more efficient widths, but higher abstention rate)

We also provide a quick flow-chart for deciding parameter values

1. **Is data stationary (IID)?**
    - **YES**: Use non-adaptive method (Section 5)
        - Set $c = d = \frac{\alpha}{2}$, $\tau = [0.03, 0.05]$
    - **NO**: Use adaptive method (Section 6)
        - Set $c = d = \frac{\alpha}{2}$, $\gamma = \eta = 0.01$

2. **Are both stages equally reliable?**
    - **YES**: Keep $c = d = \frac{\alpha}{2}$
    - **NO**: Increase quantile level for less reliable stage
        - Upstream unreliable: $c = \alpha$, $d = \frac{\alpha}{2}$
        - Downstream unreliable: $c = \frac{\alpha}{2}$, $d = \alpha$

3. **What is the cost of coverage violations?**
    - **HIGH**: $\delta = 0.1$ (accept more abstentions)
    - **MED**: $\delta = 0.3$ (default)
    - **LOW**: $\delta = 0.5$ (prioritize efficiency)

4. **How much data is available?**

- $|S_{\text{cal}}| > 200$: Use fine grid for $\Lambda$ (0.1 increments)
- $|S_{\text{cal}}| < 100$: Use coarse grid for $\Lambda$ (0.2 increments)

### D.3 NON-ADAPTIVE EXPERIMENTS

We discuss results for the non-adaptive intervals using our residual decomposition as well as hyperparameter ablation studies.

#### D.3.1 COVERAGE UNDER EASY SETTINGS

We implement both Definition 4 and Definition 3 separately and test them on simple IID data with linear relationships between $w, x, y$. We also include a results for an IID synthetic data featuring a nonlinear relationship in which the upstream task is a binary classification task of diagnosing a patient for a disease given their health (using logistic regression), and the downstream model predicts the cost of treatment given the upstream probabilistic prediction (using XGBoost). Additionally, we implement these prediction interval methods in a $\phi$-mixing setting which we provide guarantees for in Section C.2 where the data is generated by a stationary AR(1) process with bounded uniform noise (known to be a $\phi$-mixing process Athreya and Pantula (1986)) and linear relationships between $w, x, y$. We report the width and coverage of the methods in Section D.3.1.

Table 4: Average coverage and width with standard deviation across 50 runs for the scaling components method with fixed $c = d = \alpha = 0.1$ and FWER as in Definition 4, which we denote as $\text{SR}_{a,b}$, the separate quantiles method with $c = d = 0.05$ as in Definition 3, denoted as $\text{SR}_{c,d}$, SC, and WSC. We examine these under linear and non-linear relationships and non-IID data

| Shift | Metric | $\text{SR}_{a,b}$ | $\text{SR}_{c,d}$ | SC | WSC |
|---|---|---|---|---|---|
| Linear | Coverage | 0.9439±0.02 | 0.9492±0.01 | 0.8971±0.01 | 0.9062±0.02 |
| | Width | 2.3555±0.13 | 2.3930±0.07 | 1.9924±0.06 | 2.0584±0.13 |
| Non-linear | Coverage | 0.9416±0.01 | 0.9794±0.01 | 0.8969±0.01 | 0.9300±0.02 |
| | Width | 4.6343±0.14 | 5.2007±0.06 | 4.2179±0.08 | 4.5089±0.16 |
| Non-IID (linear) | Coverage | 0.9561±0.02 | 0.9886±0.01 | 0.8997±0.02 | 0.9067±0.03 |
| | Width | 3.1299±0.20 | 3.6106±0.12 | 2.6749±0.08 | 2.7304±0.15 |

We observe that both $\text{SR}_{a,b}$, $\text{SR}_{c,d}$ are conservative for these settings, producing wider intervals but attaining the desired $\alpha$ coverage guarantee. Thus, our methods tend to over-cover in easier settings such as IID data and stationary $\phi$-mixing conditions because FWER hypothesis testing requires an error rate below $\alpha$ whereas for other methods simply being close is sufficient. However, as we see in the ablation studies below, this can be easily remedied if desired by including a tolerance for the hypothesis test itself.

#### D.3.2 HYPERPARAMETER STUDY

We vary the tolerance parameter $\tau$ which alters the hypothesis test, obtaining $\text{Bin}(n, \alpha + \tau)$ to demonstrate that when comparing our method with some tolerance, the width and coverage even under the IID settings are comparable and do not over-cover. We observe that even low values of the tolerance parameter, such as $\tau = 0.03$, are sufficient to make $\text{SR}_{a,b}$ similar to the black-box methods in the IID setting. This suggests that $\text{SR}_{a,b}$ is not simply over-covering, but rather enforcing stricter criteria for valid prediction intervals than the black-box approaches. These stricter requirements likely contribute to its improved performance under distributional shifts. Therefore, if one seeks tighter but potentially less robust intervals, adjusting the tolerance parameter $\tau$ provides a principled way to trade-off between coverage robustness and interval sharpness, however experimentally, unless stated otherwise, we use $\tau = 0$.

We also vary the $\delta$ parameter, which controls the FWER rejection threshold, which for smaller values encourages more sensitivity of the method overall to abstain from producing an interval; in a sense it has a similar effect to $c, d$ parameters but collectively for both stages at once rather than a stage-wise sensitivity. Lastly, we consider different values of $\alpha$ for the IID setting, using $\tau = 0.05$ for efficiency

Table 5: Coverage and average width ($\pm$ std) across 50 runs for varying $\tau \in \{0.01, 0.03, 0.05\}$ for $\alpha = 0.1$. Only $SR_{a,b}$ depends on $\tau$, thus the other methods do not report values for each $\tau$

| Method | Metric | $\tau = 0.01$ | $\tau = 0.03$ | $\tau = 0.05$ |
|---|---|---|---|---|
| $SR_{a,b}$ | Coverage | 0.9240$\pm$0.01 | 0.9150$\pm$0.02 | 0.9019$\pm$0.02 |
| | Width | 2.1750$\pm$0.09 | 2.0922$\pm$0.08 | 2.0192$\pm$0.11 |
| $SR_{c,d}$ | Coverage | 0.9516$\pm$0.01 | – | – |
| | Width | 2.4032$\pm$0.05 | – | – |
| SC | Coverage | 0.8981$\pm$0.01 | – | – |
| | Width | 2.0051$\pm$0.04 | – | – |
| WSC | Coverage | 0.9015$\pm$0.02 | – | – |
| | Width | 2.0320$\pm$0.13 | – | – |

Table 6: Coverage and average width ($\pm$ std) across 50 runs for varying $\delta \in \{0.1, 0.3, 0.5, 0.7, 0.9\}$. Only $SR_{a,b}$ depends on $\delta$

| Method | Metric | $\delta = 0.1$ | $\delta = 0.3$ | $\delta = 0.5$ | $\delta = 0.7$ | $\delta = 0.9$ |
|---|---|---|---|---|---|---|
| $SR_{a,b}$ | Coverage | NA | NA | 0.8969$\pm$0.08 | 0.9294$\pm$0.02 | 0.9268$\pm$0.02 |
| | Width | NA | NA | 2.2231$\pm$0.09 | 2.2195$\pm$0.10 | 2.1867$\pm$0.11 |
| $SR_{c,d}$ | Coverage | 0.9504$\pm$0.01 | – | – | – | – |
| | Width | 2.3919$\pm$0.05 | – | – | – | – |
| SC | Coverage | 0.8992$\pm$0.01 | – | – | – | – |
| | Width | 1.9987$\pm$0.04 | – | – | – | – |
| WSC | Coverage | 0.9082$\pm$0.02 | – | – | – | – |
| | Width | 2.0603$\pm$0.14 | – | – | – | – |

and we see that our method is similarly calibrated to the other non-adaptive methods. We visualize this by plotting empirical coverage against empirical width across various levels of $\alpha \in [0.05, 0.15]$ in Figure 4. The opacity of each point represents the value of $\alpha$ used, the base color represents the method used, and we represent the standard deviation in coverage and width at each point via the bars extending from the point. Specifically, the efficiency of our method (blue) increases as the miscoverage rate $\alpha$ decreases, approach SC (green) in performance and being more efficient than WSC (purple). In comparison to our method with $\tau = 0$ (orange), this contrast demonstrates that reasonable values of $\tau$ allow for greatly improved efficient performance under IID settings.

### D.4 ADAPTIVE

We include additional experiments using additional synthetic experiments with covariate shift and real-world stock data to demonstrate the flexibility of our method. Furthermore, we provide visualizations of the scaling parameters to demonstrate the identification of the decomposition. In addition, we perform hyperparameter sweeps over the synthetic dataset described in Section 7. We also include comparisons to DtACI, although DtACI performs poorly due to being designed for one step ahead updates, rather than with a forecasting horizon.

#### D.4.1 STOCKS DATASET

We consider a dataset consisting of daily closing values of three S&P 500 stocks, AAPL, AMZN, MSFT with the goal of forecasting the AAPL closing value. We do so via a two-stage approach in which the first-stage is an N-BEATS forecasting model that forecasts the three stocks at a 6-day horizon, then the second-stage is ridge regression that predicts AAPL given the forecasted values to correct for error from the first stage. We rescale the data and compare against ACI, Conformal PID, and OCID methods to showcase the robustness of our method. The coverage of these methods is presented in Figure 5.

We observe that our adaptive method provides better coverage than the other three methods, particularly when the forecasts perform worse in July and October, seeing a much less dramatic drop in

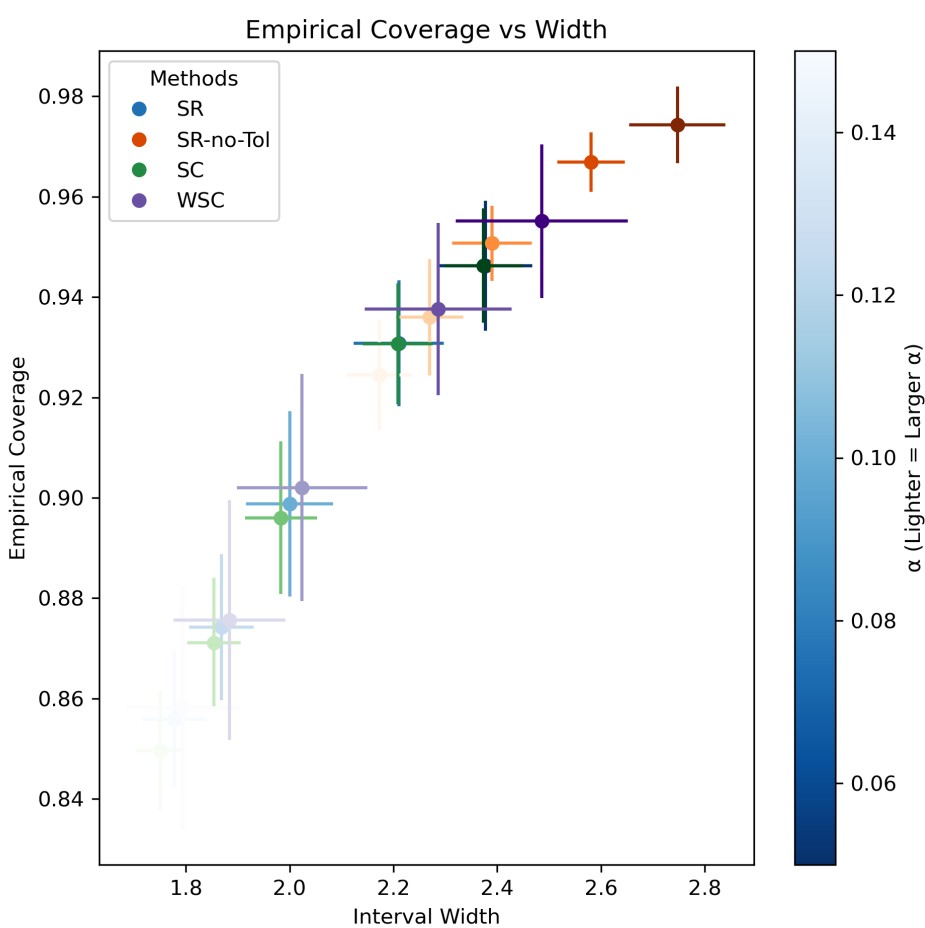

Figure 4: Empirical width vs coverage for the non-adaptive methods under IID data. Our method uses $\tau = 0.05$.

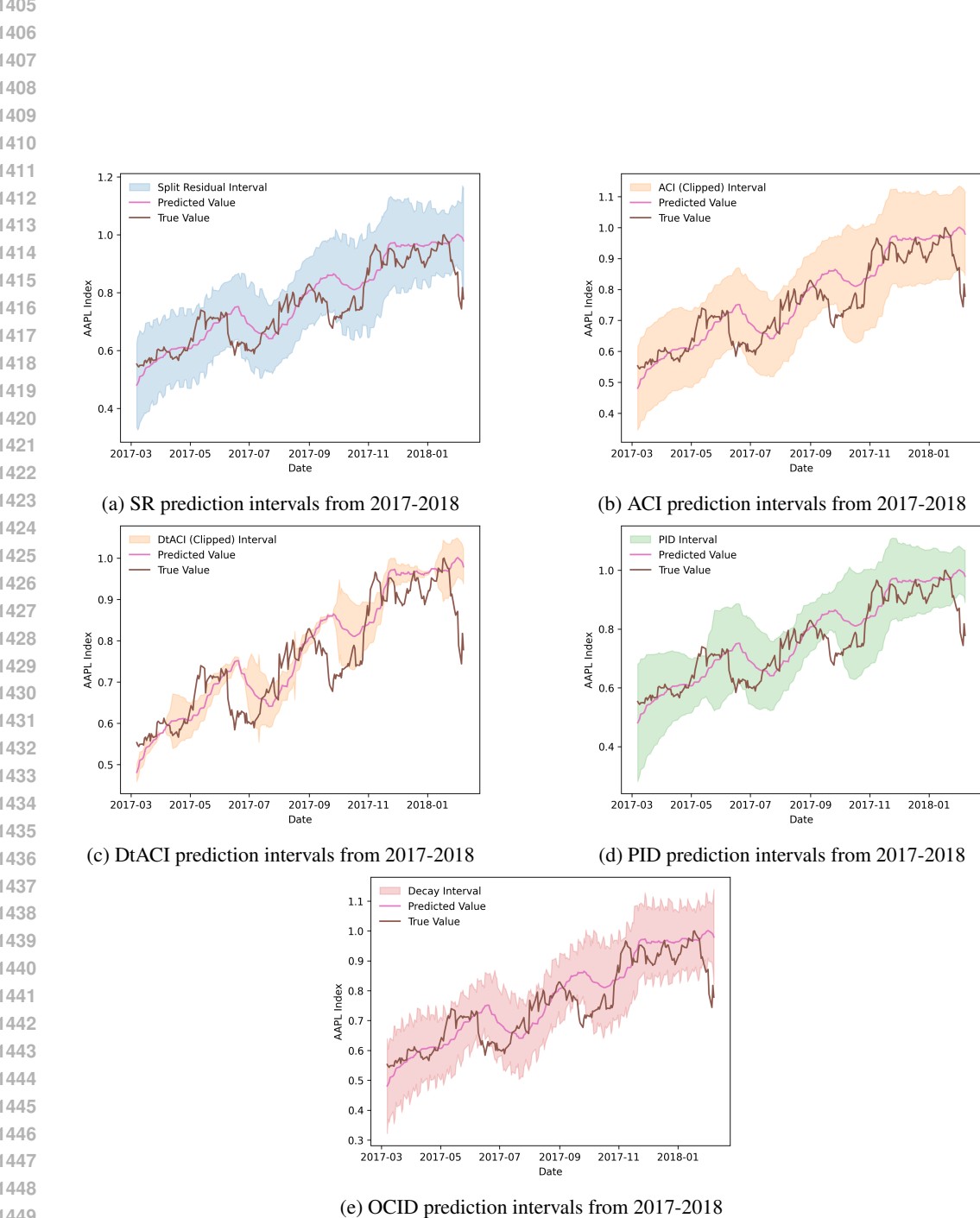

(a) SR prediction intervals from 2017-2018

(b) ACI prediction intervals from 2017-2018

(c) DtACI prediction intervals from 2017-2018

(d) PID prediction intervals from 2017-2018

(e) OCID prediction intervals from 2017-2018

Figure 5: Coverage of prediction intervals for $\alpha = 0.1$, $\delta = 0.3$, $\gamma = 0.01$, $\eta = 0.01$, $k = 24$

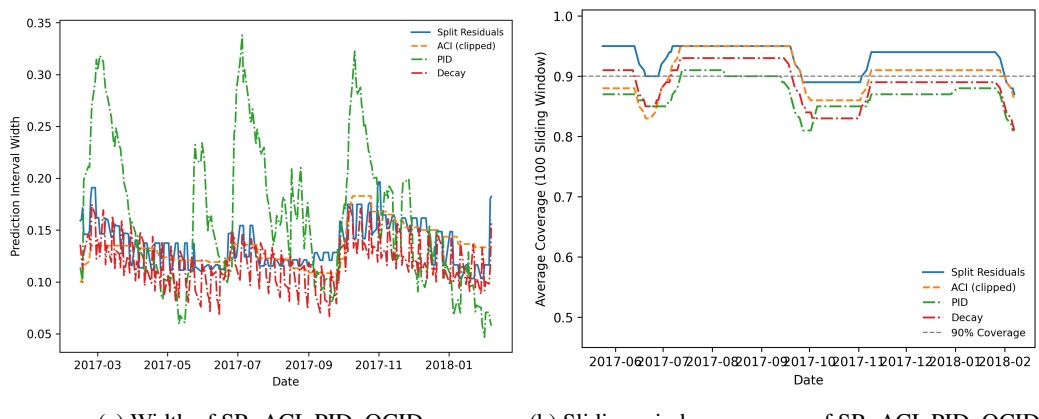

(a) Width of SR, ACI, PID, OCID    (b) Sliding window coverage of SR, ACI, PID, OCID

Figure 6: Performance of various intervals on the stocks dataset for $\alpha = 0.1$, $\delta = 0.3$, $\gamma = 0.01$, $\eta = 0.01$, $k = 24$.

performance while not being overly tight. Our method uses almost the same average width (0.1347) as ACI (0.1344), while the other two methods have tighter intervals but significantly worse coverage. We list these results in the table below Table 7 and also the sliding window coverage plots Appendix 6. We see that there are two main sudden distribution shifts, and that in comparison to ACI, our method is able to capture the second distribution shift, while OCID is also able to but has volatile intervals during the stable period prior to the distribution shifts.

Table 7: Average and standard deviation of width over time, and minimum average coverage over a sliding window of the last 100 observations for each method on the stocks dataset

| Method | Avg. Width | Std. Width | Min Coverage |
|---|---|---|---|
| SR(adaptive) | 0.1348 | 0.0216 | 0.89 |
| ACI | 0.1345 | 0.0193 | 0.86 |
| PID | 0.1183 | 0.0226 | 0.85 |
| OCID | 0.1240 | 0.0348 | 0.83 |

### D.4.2 DtACI

We also include results on the DtACI method in comparison to the others for the same synthetic dataset with two drastic distribution shifts used in Section 7.2, with the plots shown in Figure 7a. We observe that DtACI outputs values of $\alpha_t > 1$ which we clip to 1, resulting in very conservative intervals. This is most likely due to the fact that it is not designed for forecasting with a horizon and suffers in performance because of that. We see similar behavior for the automobile indicators dataset (Figure 8), in which DtACI achieves good coverage but is wider ( compared to our method, ) because it often outputs $\alpha_t = 1$. Interestingly, this does not hold for the stocks dataset, where DtACI is produces the tightest intervals but has the worst coverage.

### D.4.3 AUTOMOBILE VISUALIZATIONS

We also include additional visualizations of width and coverage for the automobile dataset Figure 9. We note that our average width over the entire range is strong in efficiency (21.085), only behind PID (18.469) and ACI (15.696) which have poor coverage, whereas OCID and DtACI are all wider (31.058, 31.356), respectively. We also include comparisons of empirical coverage and width of our method compared against ACI, PID, and OCID across various values of $\alpha \in [0.1, 0.5]$ in Figure 10. These values are averaged over the last four years of the automobile indicator data to specifically capture the COVID-19 spike. We see that our coverage is strong across multiple values of $\alpha$ while

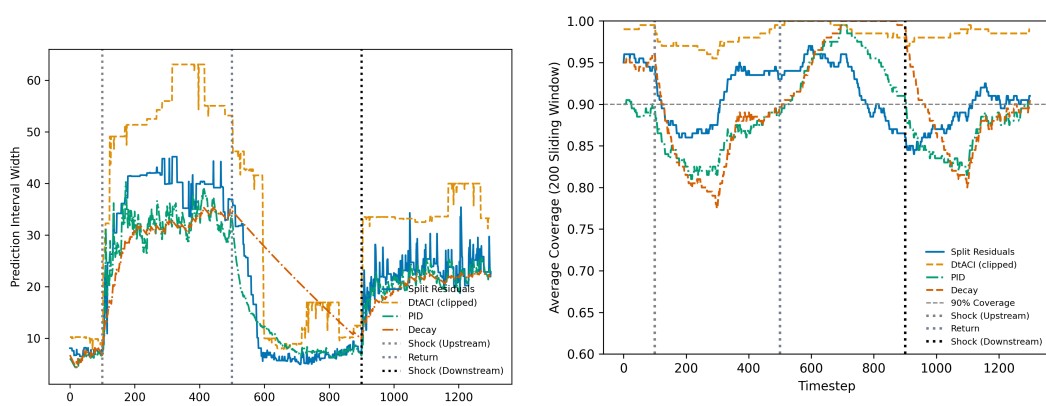

(a) Widths of the intervals for $\alpha = 0.1$, $\delta = 0.1$, $\gamma = 0.01$, $\eta = 0.01$, $k = 100$

(b) Coverage over the last 200 points. Our method remains more robust, particularly to the upstream distribution shift

Figure 7: Comparison of interval width and coverage robustness under synthetic distribution shifts with DtACI

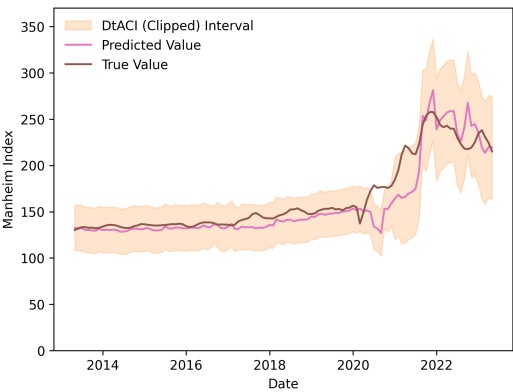

Figure 8: DtACI interval for automobile indicators dataset. It captures the jump in 2020 at the cost of wide intervals throughout

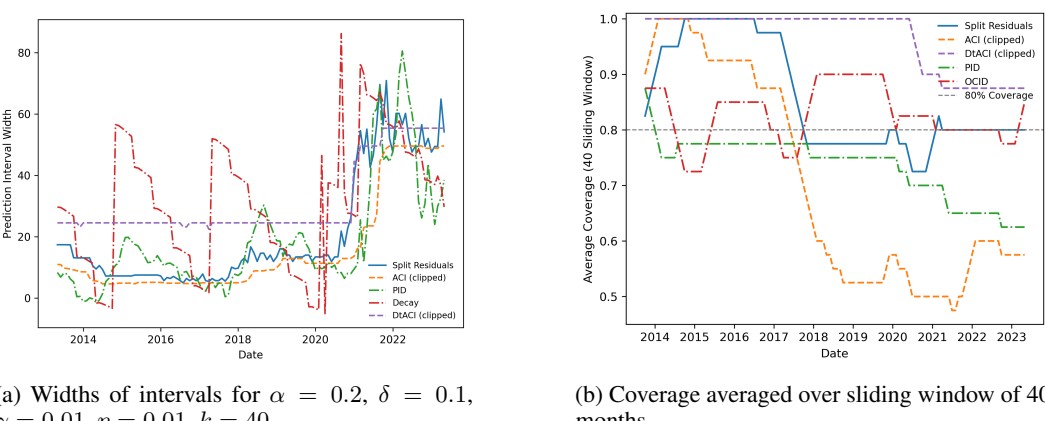

(a) Widths of intervals for $\alpha = 0.2$, $\delta = 0.1$, $\gamma = 0.01$, $\eta = 0.01$, $k = 40$

(b) Coverage averaged over sliding window of 40 months

Figure 9: Comparison of interval width and coverage for automobile indicator dataset

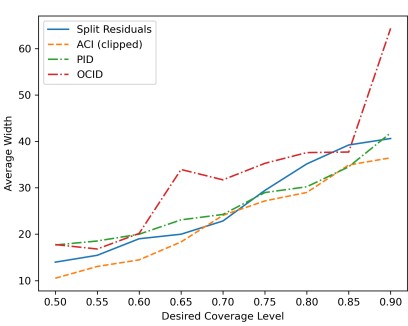 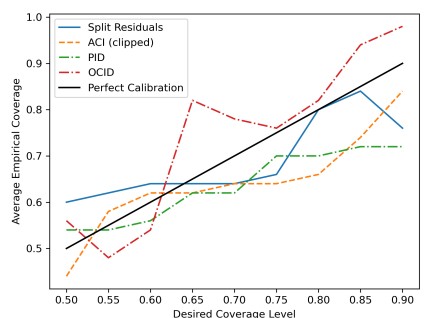

(a) x-axis: desired coverage level, y-axis: average widths of intervals for $\alpha \in [0.1, 0.5]$

(b) x-axis: desired coverage level, y-axis: average coverage of intervals

Figure 10: Comparison of interval width and coverage for automobile indicator dataset, $\delta = 0.3$, $\gamma = 0.01$, $\eta = 0.01$, $k = 40$. Averaged over last 50 months of automobile dataset

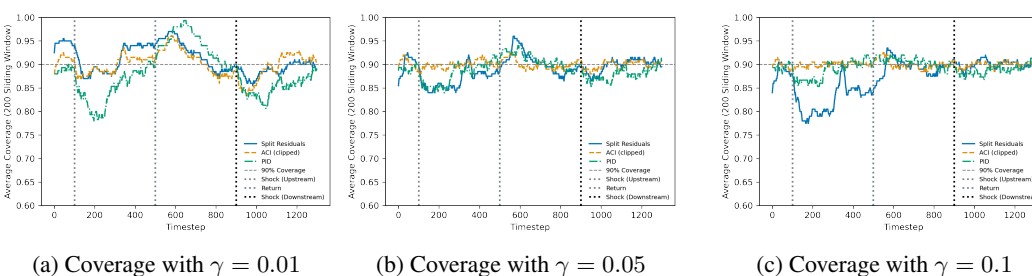

(a) Coverage with $\gamma = 0.01$     (b) Coverage with $\gamma = 0.05$     (c) Coverage with $\gamma = 0.1$

Figure 11: We use $\alpha = 0.1$, $\delta = 0.1$, $\eta = 0.01$, $k = 100$

remaining generally efficient compared to PID and OCID and well calibrated, particularly around $1 - \alpha = 0.8$. Note that coverage dips at $1 - \alpha = 0.9$ due to abstention which we do not count.

### D.4.4 HYPERPARAMETER STUDY

First, we visualize the effect of various $\gamma$ values, which affects the rate at which $\alpha_t$ changes. We find that our method excels at lower values of $\gamma = 0.01$ and performance decays at higher values such as $\gamma = 0.1$, particularly because this causes $\alpha_t$ to drop to extremely low values, making the algorithm abstain from producing intervals under large shifts. We display the coverage effects for synthetic data in Figure 11 and for Manheim data in Figure 12, where we see a similar decay as $\gamma$ increases.

ACI and PID improve slightly at higher $\gamma$ on the synthetic dataset. While one might argue that this implies that one can just use higher values of $\gamma$ without the need for the split residuals method, using a lower $\gamma$ is "safer" as it does not adjust to noisy signals as significantly. Accordingly, the widths of the other methods become extremely volatile with increased $\gamma$. Thus, our method is able to still catch significant jumps while remaining at a safer, consistent step-size due to always considering conservative scaling parameters in $\Lambda_{\text{val}}$. Furthermore, even when considering multiple values of $\gamma$, on the Manheim dataset, our method with $\gamma = 0.01$ still outperforms the other methods in coverage at all values of $\gamma$, and the performance of the other methods actually decay in coverage as $\gamma$ increases. OCID does not use same step-size parameter so it remains unaffected.

We also provide tables for other hyperparameters $\gamma, \eta, \delta, k$ in Tables 9 to 11, fixing them at default values of $0.01, 0.01, 0.3, 100$ respectively when not varying them. We see that increasing $\gamma$ has a rough impact of decreasing width, while increasing $\eta$ has a weaker widening effect, $\delta$ has no particular effect on width, but low values encourage abstention, and lastly small window lengths also encourage abstention, and generally longer windows results in slightly larger intervals.

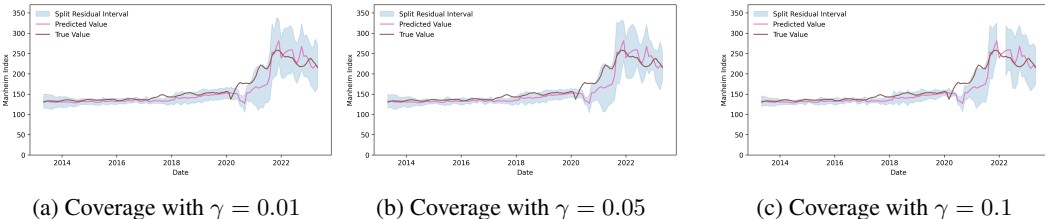

(a) Coverage with $\gamma = 0.01$    (b) Coverage with $\gamma = 0.05$    (c) Coverage with $\gamma = 0.1$

Figure 12: We use $\alpha = 0.2$, $\delta = 0.3$, $\eta = 0.01$, $k = 40$

Table 8: Coverage, width, and maximum improvements in coverage vs. baselines (ACI, PID, OCID) across 50 runs for varying $\gamma \in \{0.01, 0.03, 0.05, 0.1\}$

| **Metric** | $\gamma = 0.01$ | $\gamma = 0.03$ | $\gamma = 0.05$ | $\gamma = 0.1$ |
|---|---|---|---|---|
| Width | 22.92±0.13 | 24.75±0.10 | 21.26±0.12 | 21.93±0.12 |
| Coverage | 0.835±0.02 | 0.815±0.02 | 0.595±0.02 | 0.715±0.01 |
| Max $\Delta$Coverage vs ACI | 0.075±0.02 | 0.030±0.01 | 0.025±0.01 | 0.020±0.01 |
| Max $\Delta$Coverage vs PID | 0.075±0.02 | 0.045±0.01 | 0.045±0.01 | 0.025±0.01 |
| Max $\Delta$Coverage vs OCID | 0.130±0.03 | 0.090±0.02 | 0.110±0.02 | 0.100±0.03 |

### D.4.5 VISUALIZATION OF RESIDUAL COMPONENTS

Here, we provide extra visualizations of the $\Delta R_1$ and $R_2$ parameters along with how the weights $a, b$ change over time, for the same synthetic setup as Section 7.2 below in Figures 13 and 14a.

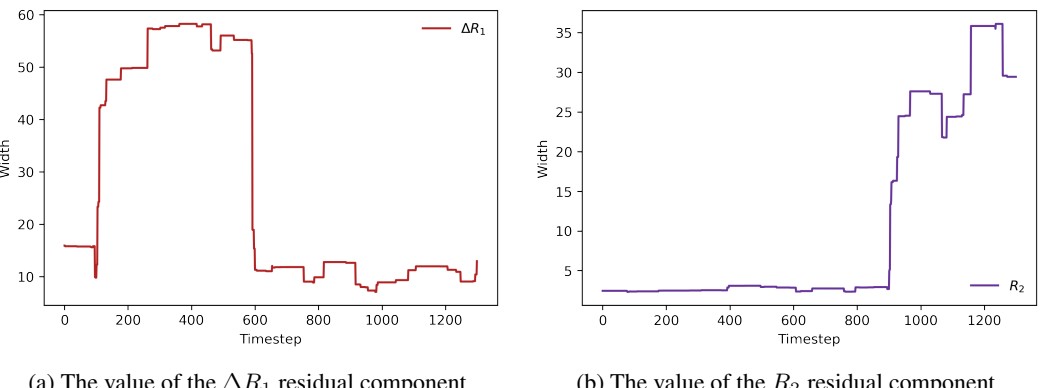

(a) The value of the $\Delta R_1$ residual component    (b) The value of the $R_2$ residual component

Figure 13: A visualization of how $\Delta R_1, R_2$ change as distribution shifts occur in the data. Note that they are taken at $1 - c_t, 1 - d_t$ quantiles respectively. We use $\alpha = 0.1$, $\delta = 0.1$, $\gamma = 0.01$, $\eta = 0.01$, $k = 100$.

We observe that when the upstream distribution shift occurs, $\Delta R_1$ captures this change, then after it returns, $\Delta R_1$ correspondingly shrinks back to the baseline level. $R_2$ correspondingly captures the second downstream distribution shift.

We also see this relationship reflected in the $a, b$ scaling parameters selected by our method as we see them adapt to the upstream distribution shift, the return, then the downstream distribution shift in a similar fashion (Figure 14a). Lastly, we plot the ratios $R/(\Delta R_1)$ and $R/R_2$ over time in Figure 14b.

We see that the shifts at $t = 100, 500, 900$ are visibly identifiable: initially $R/R_2$ is high during the upstream shift as the majority of error comes from $R_1$. When the upstream shift returns to baseline, we observe that the upstream remains the main source of error, and finally in the downstream shift, $R_2$ becomes the dominant source of error. Concretely, $\Delta R_1$ dominates between $t = 100$ to $t = 500$ when the upstream undergoes a shift and $R_2$ dominates when the downstream undergoes a shift from

Table 9: Coverage, width, and maximum improvements in coverage vs. baselines (ACI, PID, OCID) across 50 runs for varying $\eta \in \{0.01, 0.03, 0.05, 0.1\}$.

| Metric | $\eta = 0.01$ | $\eta = 0.03$ | $\eta = 0.05$ | $\eta = 0.1$ |
|---|---|---|---|---|
| Width | 23.50±0.12 | 24.22±0.13 | 24.16±0.14 | 23.96±0.10 |
| Coverage | 0.835±0.03 | 0.835±0.02 | 0.855±0.01 | 0.860±0.02 |
| Max $\Delta$Coverage vs ACI | 0.080±0.01 | 0.065±0.02 | 0.075±0.01 | 0.075±0.01 |
| Max $\Delta$Coverage vs PID | 0.105±0.03 | 0.090±0.01 | 0.090±0.02 | 0.075±0.01 |
| Max $\Delta$Coverage vs OCID | 0.125±0.02 | 0.135±0.01 | 0.115±0.03 | 0.115±0.02 |

Table 10: Coverage, width, and maximum improvements in coverage vs. baselines (ACI, PID, OCID) across 50 runs for varying $\delta \in \{0.05, 0.1, 0.3, 0.5, 0.7\}$

| Metric | $\delta = 0.05$ | $\delta = 0.1$ | $\delta = 0.3$ | $\delta = 0.5$ | $\delta = 0.7$ |
|---|---|---|---|---|---|
| Width | NA | 23.30±0.12 | 22.70±0.13 | 24.11±0.09 | 23.20±0.11 |
| Coverage | NA | 0.865±0.03 | 0.840±0.02 | 0.835±0.01 | 0.845±0.02 |
| Max $\Delta$Coverage vs ACI | NA | 0.075±0.01 | 0.065±0.02 | 0.055±0.01 | 0.065±0.02 |
| Max $\Delta$Coverage vs PID | NA | 0.080±0.01 | 0.100±0.02 | 0.080±0.01 | 0.070±0.01 |
| Max $\Delta$Coverage vs OCID | NA | 0.115±0.02 | 0.130±0.02 | 0.110±0.03 | 0.105±0.03 |

$t = 900$ onward. Thus we can observe which stage is responsible for the error with extreme clarity allowing diagnostic retraining. In particular, if we retrain at $t = 200$ after the first upstream shift, we can choose to either retrain just $\hat{\mu}_1$ or both $\hat{\mu}_1$ and $\hat{\mu}_2$. We observe that just retraining $\hat{\mu}_1$ results in a width decrease of 24.6439 for the SR method, while retraining both models, results in a decrease of 25.3451, demonstrating that it is sufficient for our framework to only retrain the affected upstream.

We similarly plot the used scaling parameters $(a, b)$ over time for the Manheim used vehicles dataset, and observe that there is in fact an asymmetry in the scaling factors in relation to the shifts occurring during 2018-2020. Likewise, we also observe asymmetry in the stocks dataset, with more emphasis on the downstream. This gives us an idea of which part of the model fails and how the shift is affecting it (Figure 15).

### D.4.6 WITHOUT QUANTILE LEVEL ADJUSTMENTS

In this experiment, we verify that the $c_t, d_t$ heuristic update step of the algorithm is necessary for improved performance. Using the experimental setup with upstream and downstream distribution shifts as Section 7.2, we consider the case in which we fix $c_t, d_t$ to initial values 0.01 $\forall t$, in Figure 16. We observe that fixing $c, d$ makes the algorithm unable to adapt properly to shifts as it produces $\Lambda_{\text{val}}$ sets frequently after the initial upstream distribution shift, which lowers average coverage (considering that to be a miscoverage). Thus, we see that adjusting $c_t, d_t$ allows us to mitigate abstention, however, if desiring more sensitivity to shifts, keeping them fixed at larger values could serve as a diagnostic for retraining, similar to the non-adaptive case (Table 1).

### D.4.7 COVARIATE SHIFT

We consider covariate shift, in which the distribution of $w$ suddenly changes, at $t = 100, 500, 900$ which affects the entire pipeline. This shift does not have a specific upstream/downstream affect, but our method is still able to perform comparably well to the ACI, PID, and OCID methods Figure 17. However, all the methods perform similarly and it is difficult to distinguish them. Thus, while our method provides clear differentiation between upstream/downstream shifts, it is still able to adapt to other types of shifts without overcompensating width in comparison to other methods. However, it does not obtain any explicit advantage compared to the other methods and is still wider on average; thus we see that our method excels under asymmetric stage-wise shifts, and furthermore comparatively performs even better under upstream shifts.

Table 11: Coverage, width, and maximum improvements in coverage vs. baselines (ACI, PID, OCID) across 50 runs for varying window length $k \in \{10, 50, 100, 150\}$

| Metric | $k = 10$ | $k = 50$ | $k = 100$ | $k = 150$ |
|---|---|---|---|---|
| Width | NA | 22.21±0.08 | 23.24±0.07 | 24.32±0.13 |
| Coverage | NA | 0.855±0.02 | 0.856±0.03 | 0.835±0.01 |
| Max $\Delta$Coverage vs ACI | NA | 0.050±0.01 | 0.070±0.02 | 0.070±0.01 |
| Max $\Delta$Coverage vs PID | NA | 0.080±0.01 | 0.090±0.02 | 0.080±0.01 |
| Max $\Delta$Coverage vs OCID | NA | 0.115±0.01 | 0.100±0.01 | 0.120±0.01 |

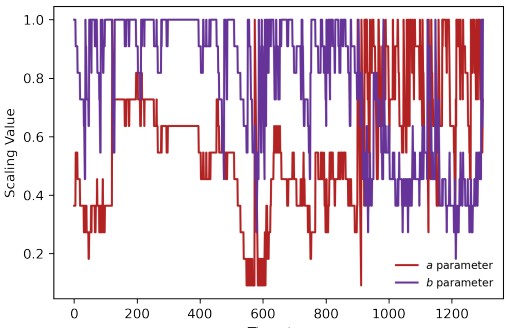

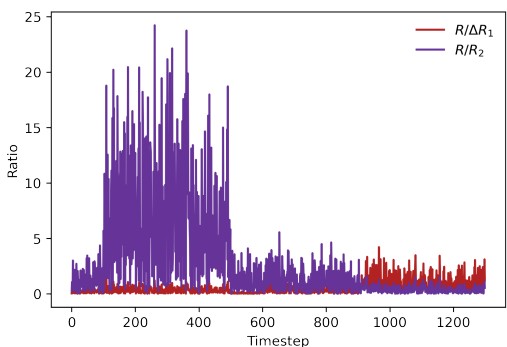

(a) Visualization of scaling parameters $a, b$ over time.

(b) Ratio of $R$ to $\Delta R_1$ and $R_2$ respectively

Figure 14: Plots of residual component relations over time on the synthetic dataset. We use $\alpha = 0.1$, $\delta = 0.1$, $\gamma = 0.01$, $\eta = 0.01$, $k = 100$

### D.4.8 SINGLE-STAGE BASELINE

We consider a simple baseline in which the upstream stage is removed entirely, fitting directly from upstream input $w$ to downstream output $y$. We consider this baseline for both the synthetic distribution shift data and automobile indicator data. We apply the baseline adaptive methods (ACI, PID, OCID) to determine if the improvement of our method stems from the improved predictive power of the two-stage model or from exploiting the error decomposition.

**Synthetic data with sudden shifts.** Instead of two stages of least squares, we consider a model that fits only one least squares model directly from $w$ to $y$ (Figure 18). We observe the performance of ACI, PID, OCID on this single-stage model and observe that while the widths are slightly smaller, the coverage performance is similar between single-stage and two-stage and still worse than our method, which demonstrates that the two-stage predictor does not necessarily improve coverage performance. Thus our method, which does have improved performance in the two-stage regime, is not merely benefiting from a better predictor, but leveraging the two-stage structure. Similar patterns hold for the real-world dataset as we see below.

**Automobile Indicators.** Instead of using our two-stage pipeline, we directly fit a single-stage predictive model to forecast the Manheim index, and then apply each of the baseline adaptive conformal methods (ACI, PID, OCID). Figures 19 and 20 report interval widths, empirical coverage, and example prediction intervals when the single-stage model is VAR(3) and ARIMA(2,0,1) respectively.

Unsurprisingly, the single-stage predictive models perform substantially worse than the two-stage model. ARIMA largely produces a lagged version of the truth, while VAR mispredicts the 2020 spike, getting both the direction and timing wrong. By contrast, our two-stage pipeline "catches up" to the spike and produces a substantially more accurate forecast. In terms of coverage, ACI and PID still fail to capture the 2020 COVID distribution shift. Furthermore, the coverage gap remains similar between single and two-stage. For example, ACI and PID end with coverage of 0.53,0.55 (ARIMA) or 0.61, 0.61 (VAR) respectively, compared to 0.58, 0.6 with the two-stage model, still well below

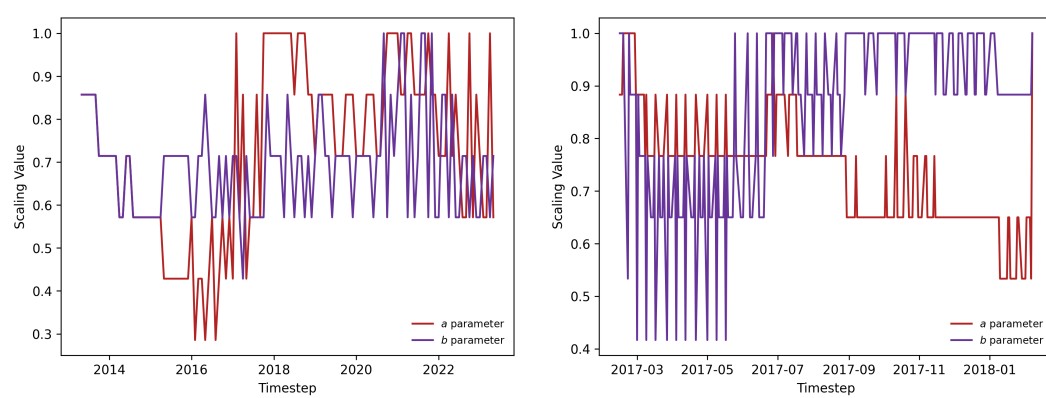

(a) Values of $(a, b)$ over time for the automobile dataset

(b) Values of $(a, b)$ over time for the stocks dataset

Figure 15: A visualization of how $(a, b)$ change as distribution shifts occur in real-world data

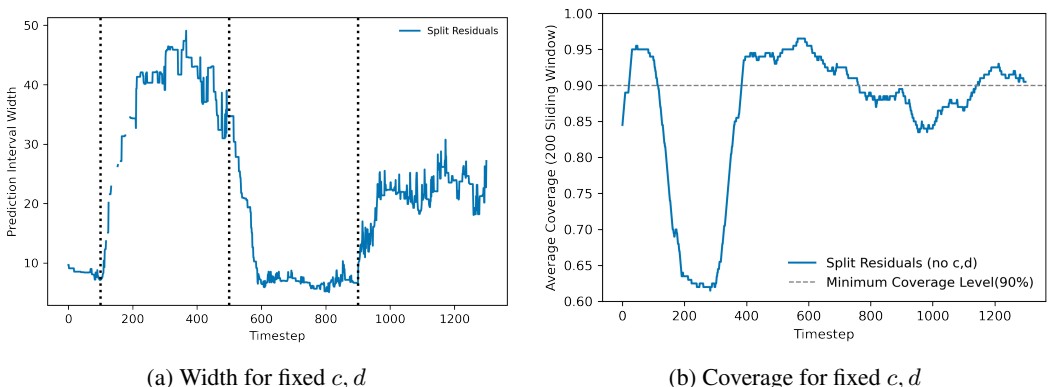

(a) Width for fixed $c, d$

(b) Coverage for fixed $c, d$

Figure 16: A visualization of how freezing $c_t, d_t$ to fixed values with no updates affects performance. We use $\alpha = 0.1$, $\delta = 0.1$, $\gamma = 0.01$, $\eta = 0.00$, $k = 100$. The first distribution shift leads to abstention

desired 0.8. OCID fares better but is inefficient, with slightly worse performance on single-stage 0.75 (ARIMA), 0.71(VAR) compared to 0.84 on the two-stage model.

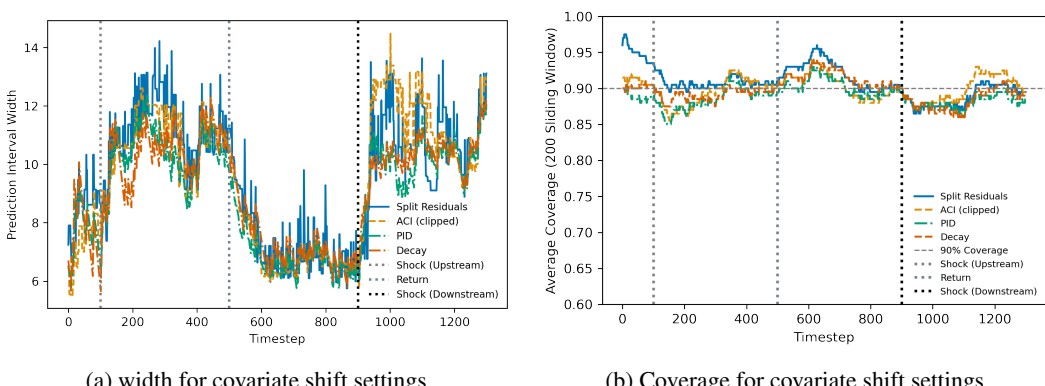

(a) width for covariate shift settings

(b) Coverage for covariate shift settings

Figure 17: The distribution of $w$ shifts from $\mathcal{N}(0, 1)$ to $\mathcal{N}(3, 2)$, then back, then $\mathcal{N}(-3, 2)$. We use $\alpha = 0.1$, $\delta = 0.1$, $\gamma = 0.01$, $\eta = 0.01$, $k = 100$

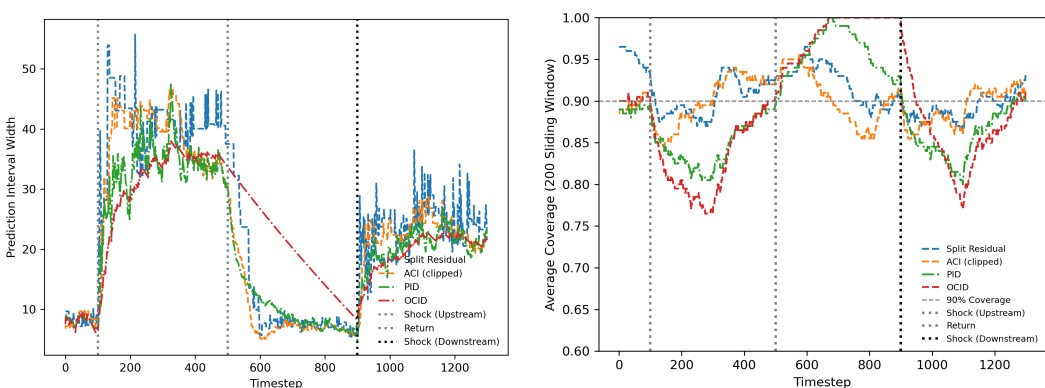

(a) Width for least squares single-stage on synthetic data

(b) Coverage for least squares single-stage on synthetic data

Figure 18: Performance of baseline methods (ACI, PID, OCID) on single-stage least squares predictor for synthetic dataset with sudden stage-wise shifts

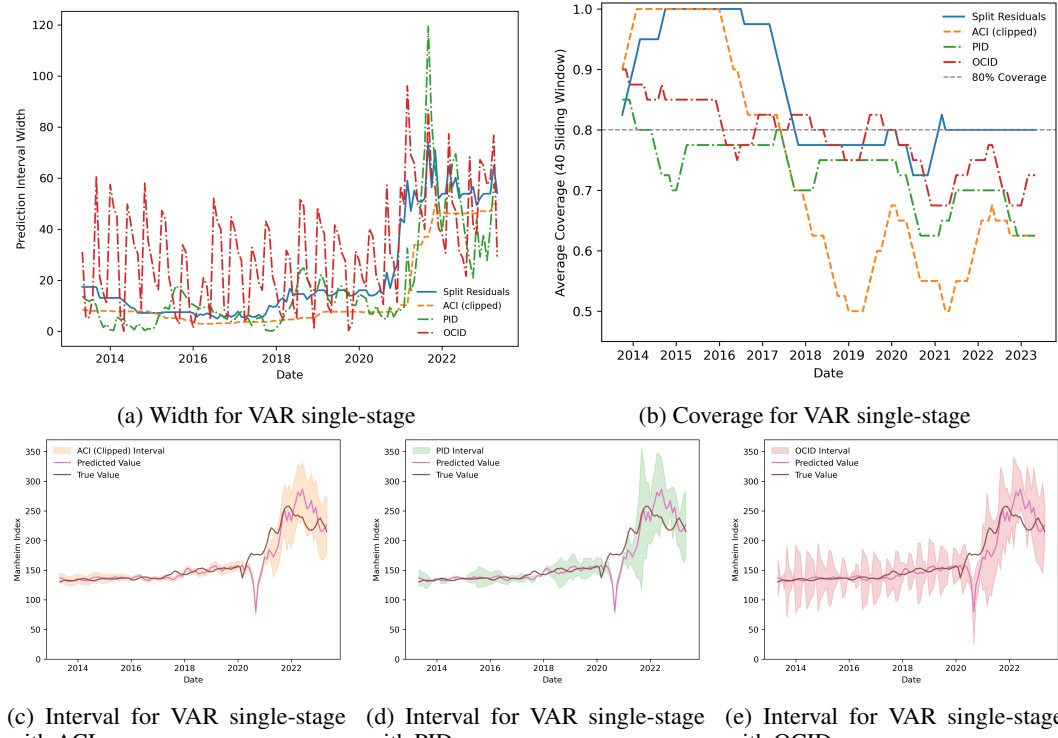

(a) Width for VAR single-stage

(b) Coverage for VAR single-stage

(c) Interval for VAR single-stage with ACI

(d) Interval for VAR single-stage with PID

(e) Interval for VAR single-stage with OCID

Figure 19: Performance of baseline methods (ACI, PID, OCID) on single-stage VAR predictor for automobile indicator dataset

However, this experiment illustrates three important points. (i) The two-stage pipeline provides clear predictive benefits over standard single-stage time-series models on this dataset. (ii) The baseline adaptive conformal methods struggle in this setting regardless of whether the predictor is single-stage or two-stage. (iii) Taken together, these results show that the improved prediction accuracy of the two-stage model alone is insufficient to obtain valid or efficient adaptive coverage: all baseline conformal methods continue to suffer from instability and undercoverage even when the predictor is strengthened. The gains achieved by our method therefore stem not merely from improved forecasting, but from explicitly exploiting the structure of the two-stage pipeline.

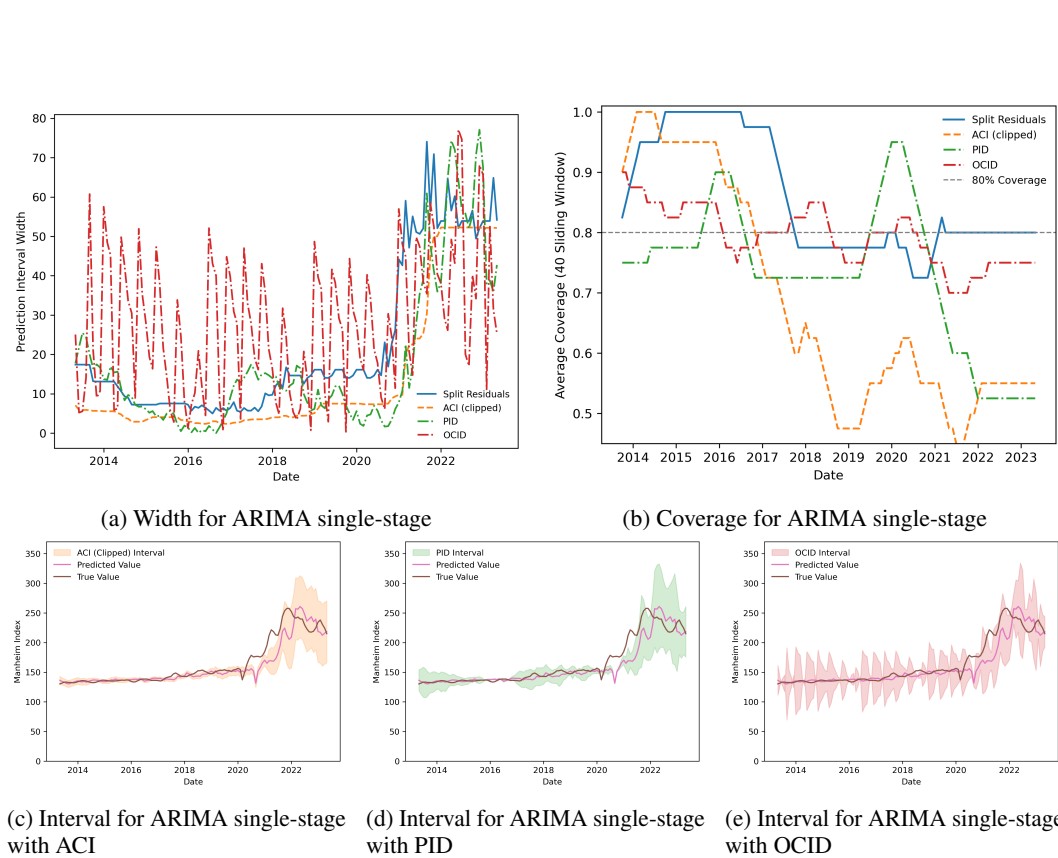

(a) Width for ARIMA single-stage

(b) Coverage for ARIMA single-stage

(c) Interval for ARIMA single-stage with ACI

(d) Interval for ARIMA single-stage with PID

(e) Interval for ARIMA single-stage with OCID

Figure 20: Performance of baseline methods (ACI, PID, OCID) on single-stage ARIMA predictor for automobile indicator dataset

