# OpenReview forum: "Modular and Adaptive Conformal Prediction for Sequential Models via Residual Decomposition"
_ICLR.cc/2026/Conference — ICLR 2026 Conference Desk Rejected Submission_

### Official Review · Reviewer_SdKo · 2025-10-21

**Soundness:** 4
**Presentation:** 4
**Contribution:** 3
**Rating:** 8
**Confidence:** 3

**Summary:**

In many cases, prediction is undertaken as a multi-stage task. While naively leveraging conformal prediction on the end-to-end prediction does yield valid uncertainty bounds, doing so forgoes the granularity of knowing the decomposition of the source of the errors. That is, for a two-stage pipeline, for instance, one cannot determine how much of the resulting uncertainty is the result of the upstream model and how much is from the downstream model. This paper proposes an approach to determine this decomposition and do so in a way that the two bounds can be combined to yield an informative, valid upper bound. In particular, they propose an automated search procedure that automatically tunes the selection of parameters to yield maximally informative prediction regions while also controlling for risk. They finally demonstrate how the method empirically yields improvements across a number of experiments.

**Strengths:**

The paper clearly identifies a flaw in existing conformal approaches for uncertainty estimation for multi-stage pipelines. The proposed solution is elegantly presented with the gradual build-up from the initially posited version where the parameters combining the quantiles of the two separate stages are fixed to the version with the automatic tuning of such parameters for optimal parameter selection to the version that similarly does auto-tuning but instead for online adaptation. The theorems are also clearly stated and well motivated by the overall storyline, with the proofs looking sound.

**Weaknesses:**

The main weakness is in the experimental validation: of the three main experiments, two were synthetically generated circumstances. Nonetheless, the comparisons to alternate methods of adaptation appear fairly comprehensive: including another “real” dataset would strengthen the submission, however.

**Questions:**

None other than: how does the model stack up in other real-world, time series data settings?

---

> ### Author Response · Authors · 2025-11-20
>
> Thank you for your thoughtful and encouraging review. We are delighted that you found our gradual build-up from the fixed parameters to adaptive methods compelling.
>
> In response to your suggestion for **additional real-world experiments**, we included an experiment on Stocks data in Appendix D.4.1 (line 1360) using observed (scaled) stock values of AAPL, MSFT, and AMZN to forecast future values of AAPL. We use a more complex NBEATS model upstream due to greater availability of data. We observe two larger distribution shifts in July and October of 2017 and that our method is able to capture the second shift where other methods do not. We observe that our method has the least degradation in empirical coverage (0.89 vs ACI 0.86, PID 0.85, OCID 0.83). Our method also is competitive in width being within a standard deviation of all the others (0.135 vs ACI 0.134, PID 0.118, OCID 0.124), where the latter two are tighter, but far below the desired coverage of 0.9. While this dataset doesn't exhibit the dramatic stage-specific shifts seen in our COVID-impacted automobile data, it demonstrates that our method provides value even in "routine" forecasting scenarios through consistent coverage improvements and stage-aware weighting.
>
> We have also done extensive additional experiments in response to other reviewers, such as comparisons to single-stage alternatives in Appendix D.4.8 which provide evidence that the improvements of our method stem from the stage-wise weighting scheme rather than the improved forecasting of the two-stage model.
>
> We deeply appreciate your recognition of our contribution's soundness and presentation clarity. The additional real-world validation further demonstrates the practical value of stage-wise uncertainty decomposition. We hope these additions strengthen what you already identified as a valuable contribution to the conformal prediction literature.

---

> > ### Comment · Reviewer_SdKo · 2025-11-24
> >
> > I thank the authors for their rebuttal. My question has been answered, and I stand by my score.

---

> > > ### Author Response · Authors · 2025-11-25
> > >
> > > Thank you for your thoughtful review and for maintaining your positive assessment of our work. We appreciate your feedback and are glad that the contributions and clarity of the paper resonated with you.

---

### Official Review · Reviewer_D8zB · 2025-10-23

**Soundness:** 1
**Presentation:** 3
**Contribution:** 2
**Rating:** 2
**Confidence:** 5

**Summary:**

The paper is clearly written and the modular framing is easy to follow. The Figures illustrate the main ideas clearly.

Problem context: Existing conformal prediction (CP) methods typically treat the entire prediction pipeline as a black box, ignoring modular structures such as two-stage sequential models where upstream representations feed downstream predictors. The paper aims to design a stage-aware CP framework.

Motivation: Distribution shifts often affect pipeline stages asymmetrically, e.g., upstream sensors may drift while downstream mappings remain stable. Standard conformal methods cannot localize such effects, forcing full retraining. The proposed stage-wise abstraction aims to isolate these shifts by embedding stage-specific uncertainty into conformal prediction. This is conceptually relevant.

Paper's proposal: The authors introduce a decomposition of the overall residual into an upstream and a downstream component, claiming this enables uncertainty attribution to specific stages. They further propose a risk-controlled parameter selection procedure using family-wise error rate (FWER) control, and an adaptive variant for non-stationary or distribution-shifted settings.

Methodology: Intervals are constructed by summing or scaling quantiles of the decomposed residuals. FWER is used to select scaling parameters over a validation grid, and an adaptive version updates parameters over time. The paper asserts that this yields interpretable, stage-wise uncertainty attribution and robust coverage under shift.

Experiments under simulated and real-world distribution shifts are presented.

**Strengths:**

1. Paper raises a practically relevant issue of sequential/modular nature of ML frameworks and lack of stage-aware CP frameworks and uncertainty attribution methods.

2. Despite dense notation, the paper is generally well-structured and easy to follow, with experiments and diagrams that communicate the intended intuition clearly.

3. The residual decomposition could serve as a heuristic diagnostic tool to identify which stage of a pipeline contributes most to predictive uncertainty.

4. In particular, adaptive risk control with residual decomposition is practically appealing, as demonstrated through real-data experiments.

**Weaknesses:**

1. The proof of the Proposition 2 seems incorrect. It relies on the intermediate value theorem, and by noting that $(a = 0,b = 0)$ attains coverage of $0$, while $(a = 1,b = 1)$ attains coverage of $1 - 2\alpha$. Both these numbers are strictly less than $1 - \alpha$. So, intermediate value theorem doesn't apply to say that there exist $(a^{\*}, b^{\*})$ which attains coverage of $1 - \alpha$.

2. The paper claims to address non-exchangeable or distribution-shifted data, yet all coverage results (Theorem 1, 2, and Corollaries) are proved under classical exchangeability/IID. The adaptive and FWER schemes depend entirely on IID calibration samples. This is a fatal conceptual inconsistency: the method cannot claim robustness to shift when its only theoretical foundation is exchangeable-CP validity.

3. In Lines 104-120, the authors explicitly assume exchangeability of train, calibration, and test points, as well as distribution shift. This self-contradiction invalidates much of the paper’s stated novelty and confuses the reader about what regime the guarantees actually cover. If points are exchangeable, there is no shift; if there is a shift, the proofs no longer hold. This contradiction makes the entire problem formulation incoherent.

4. Experiments are small-scale, omit modern baselines which target distribution shifts such as CQR, and do not discuss computational costs, especially, when using FWER framework.


----

While the modular decomposition idea is intuitively appealing and could serve as a useful diagnostic tool for stage-wise uncertainty assessment, the paper’s theoretical contributions are weak and often inconsistent with its claims. All guarantees rest on exchangeability, yet the empirical studies focus on non-exchangeable, distribution-shifted settings where coverage consistently falls below the nominal level. Consequently, the central claim of robustness to shift is substantially overstated. The method’s true value lies in its potential practical interpretability, it could help practitioners identify and retrain unstable components of a multi-stage model, rather than in offering new theoretical insight or provable robustness.

Overall, the incorrect proof of Proposition 2, contradictions between assumptions and stated goals, and the lack of convincing empirical evidence, leads to low score.

**Questions:**

1. In Lines 481-482, The paper states that the “method maintains coverage under shifts”, yet all reported results show coverage well below the nominal level (e.g., 0.80-0.84 for a 0.90 target in Table 1). Shouldn’t the claim be that the method fails less severely than others rather than maintaining coverage? Please clarify this wording and the intended meaning of “maintains”.

2. Sections are mislabeled (Appendix A.1 claims to prove Proposition 1 but actually proves Proposition 2).

3. In Line 219, The sentence reads “we observe that `both can methods’ yield…”. This seems to be a typographical error; please correct.

4. In case of Automobile indicators dataset, $\delta$ is chosen to be $0.3$ appears unusually large for a probability bound. What motivates this choice? How sensitive are the results to smaller values $\delta = 0.1, 0.05$? Reporting these would help assess robustness.

5. The name “Candès” is inconsistently rendered as “Candes” in several places. Please standardize the spelling throughout.

---

> ### Author Response · Authors · 2025-11-20
>
> Thank you for your detailed and careful review. Your attention has identified important clarifications needed in our presentation.
>
> **Proposition 2: Typo Not Fundamental Error**
>
> We appreciate you catching the typo in Proposition 2. The proof is correct with the statement which should read: "desired coverage $1-\alpha$ ... with quantile level $\alpha/2$ for the interval $ \hat{C}_{\alpha/2, a ^ *, b ^ *} $. Thus at $a=b=0$, coverage is 0 and $a= b= 1$ with quantile level $1-\alpha / 2$ gives coverage $\geq 1- \alpha$ (not $1- 2 \alpha$ as mistyped). This update aligns with our experimental usage, where $ c = d \leq \alpha/2 $, consistent with the corrected proposition.
>
> **Theory vs Empirical Robustness**
>
> You correctly identify that exchangeability and distribution shifts is contradictory. We never intended for them to be assumed simultaneously (at in line 104). We should have been clearer about the paper's structure:
>
> Theoretical Framework (Section 4, 5):
> - Assume exchangeability for formal guarantees and proves coverage with these assumptions.
> - Never claims theoretical coverage under shifts.
>
> Empirical Motivation and Validation (Section 1, 7):
> - Distribution shifts that affect multi-stage pipelines motivate why we need a stage-aware interpretable method.
> - Experiments demonstrate empirical robustness over baselines.
> - Stage-wise decomposition helps even when theory does not guarantee it.
>
> **Beyond Exchangeability**:
> We do provide some theoretical extensions:
> - Long-run coverage (Proposition 3, Appendix A.1.1) holds without exchangeability
> - $\phi$-mixing processes (Appendix C.2) extends the FWER approach to dependent data
> - Weighted quantiles (Appendix A.3, Corollary 3) uses weighted quantiles to handle non-exchangeable settings (with a penalty).
>
> The key takeaway is that our decomposition provides diagnostic value that helps practitioners identify failing model stages even when coverage degrades, which our method is more resistant to compared to baselines.
>
> **Coverage Below Nominal**
> You are correct that 0.80 coverage with nominal 0.90 seems like failure, but consider the context that other methods are even further from 0.90 and our method conveys diagnostic information of which stage contributes more to error. Also, our coverage gains are stronger in the adaptive setting. We have revised the manuscript to say "degrades less severely" rather than "maintains coverage" to be more clear.
>
> **Modern Baseline**:
> Regarding CQR, you are correct that it is an important contribution from recent times (2019), but it addresses a slightly different problem. CQR handles local heteroskedasticity in exchangeable settings while our focus is adaptive methods for time-varying stage-wise shifts. For this reason, we compared our method against modern adaptive and shift-focused baselines, including ACI (2021), PID (2023), OCID (2024), and DtACI(2024). That said, we agree CQR would strengthen non-adaptive baselines and have done some experiments. However it has poor performance (coverage of 0.65 for nominal 0.9) but we aim to add it in revision. We suspect this is due to the quantile regressor, which maps upstream inputs directly to downstream quantiles, making it a poor choice for multi-stage pipelines.
>
> **Additional Questions**
>
> **Computational Costs**: If the calibration set size is $l$ and the grid size for $\Lambda$ is $v$, it takes $O(v^2 l)$ to test all potential $(a,b)$ pairs. In practice, we find this to take less than a second (comparable to PID) and not a bottleneck.
>
> **Scale**: Our experiments span synthetic data, 10+ years of economic indicators, and years of daily stocks data. While not massive scale, they demonstrate practical applicability.
>
> **Question about $\delta$:** We chose $\delta =0.3$ as a default so that the $\Lambda_{val}$ set is not too restricted. However, we also tested $\delta=0.1$ and obtained the same results and have updated $\delta$ to be 0.1 in the manuscript. We did an ablation study on $\delta$ (table 10, Appendix D.4.4): for smaller $\delta < 0.1$, the method abstains more often, so unless one wants a sensitive model, we recommend $\delta \geq 0.1$, with $0.3$ being a moderate choice.
>
> Finally, thank you for pointing out detailed corrections; we have made the changes.
>
> In summary, despite the technical limitations you have identified, our contribution remains valuable. We introduce the novel decomposition which leads to diagnostic identification of stage-wise error, and we show empirical coverage improvements under shifts. The goal of this paper is not solving distribution shift theoretically (an open problem) but providing practical tools for understanding and adapting to shifts via structural decomposition. We hope this clarifies that the perceived contradictions are actually intentional distinctions between theoretical foundations and empirical applications, and the typo in Proposition 2, while regrettable, does not invalidate our core contributions.

---

> > ### Comment · Reviewer_D8zB · 2025-11-24
> >
> > Thank you for the clarification on Proposition 2. The argument seems correct to me now.
> >
> > A minor suggestion to make the paper's structure more clearer: the list of assumptions should be made section-specific so readers can clearly see which assumptions apply to which results.
> >
> > Your response also confirms that the theoretical results (which rely on exchangeability) and the empirical setting (which focuses on distribution shifts) operate in fundamentally different regimes. This makes it unclear how the theoretical section supports the paper’s stated motivation. If the main goal is to detect distribution shifts in sequential pipelines, then results proved only under exchangeability do not inform the regime of interest.
> >
> > As such, without any provable guarantees under shift, the method functions primarily as a heuristic diagnostic tool. The interpretability of the stage-wise decomposition is indeed valuable, and I agree it can help practitioners identify unstable components of a pipeline. However, interpretability alone does not substitute for reliability, and empirical improvements over baselines shown in the paper do not establish generalizability.
> >
> > On Section 7.1 and the diagnostic use of abstention: the idea that choosing a larger $c$ for detecting an upstream shift, and interpreting abstention as evidence of such a shift, is interesting. However, this behavior appears to depend heavily on the user-chosen $(c,d)$ parameters. Proposition 2 suggests that $c,d \le \alpha/2$, yet in practice a practitioner might consider many combinations. Should a user actively search for $(c,d)$ pairs that lead to abstention when the FWER procedure fails to identify $(a,b)$? For instance, outside the choices reported in Table 1, what would a user expect when choosing $(c = 0.01, d = 0.03)$ if abstention?
> >
> >
> > For these reasons, I am keeping my score.

---

> > > ### Author Response · Authors · 2025-11-25
> > >
> > > Thank you for your constructive feedback on the section-specific assumptions, which we will incorporate, and for acknowledging the clarification of Proposition 2.
> > >
> > > We would like to clarify the characterization that our theory and experiments operate in “fundamentally different regimes.” As previously mentioned, we have results that go beyond exchangeability:
> > > - Appendix A.1.1, Proposition 3 (long run coverage): Our adaptive method achieves long run average coverage approaching $1-\alpha$ under arbitrary sequences, the same guarantee as Gibbs and Candès (2021), a foundational result in adaptive conformal methods.
> > > - Appendix A.2, Corollary 5 ($\phi$-mixing): We extend our FWER process/Theorem 2 to stationary $\phi$-mixing processes, providing coverage guarantees under temporal dependence—a relevant form of non-exchangeability in sequential settings.
> > > -  Appendix A.3, Corollary 3 (weighted quantiles): Using weighted quantiles on residual components, we provide coverage guarantees under Total Variation distance penalties.
> > >
> > > Furthermore, the claim that results "do not inform the regime of interest" applies equally to all conformal prediction methods (ACI, PID, OCID), none of which provide coverage guarantees under arbitrary shift.
> > >
> > > Our contribution is not to solve this open theoretical problem but to provide a framework for two-stage models that degrades more gracefully than black-box methods under shift (empirically demonstrated) and diagnostic transparency that black-box models cannot provide.
> > >
> > > Regarding interpretability "does not substitute for reliability", we note that interpretability is necessary for trustworthiness or reliability. For example, in supply chain forecasting or medical diagnosis, knowing which component is failing or contributing more to error is critically important. A black-box method that silently produces unreliable intervals can lead to dangerous outcomes.
> > >
> > > Moreover, our method achieves higher coverage empirically than baselines across a breadth of settings (upstream shifts, downstream shifts, covariate shifts, and real-world supply chain and stock market data), supporting its reliability. While no finite set of experiments ensures universal generalizability, this variety of settings supports the versatility and practical performance of our method.
> > >
> > > Finally, regarding the $c,d$ parameters, we recommend a balanced default $c = d \leq  \alpha/2$ (balanced). If the user wants to be more sensitive to the upstream, they should choose a larger $c$ and if downstream-sensitive, they should choose larger $d$. If the user chooses $c=0.01,d=0.03$ and observes abstention, then this suggests downstream failure. If they explore multiple $c,d$ choices then the pattern of abstention provides more diagnostic information—a bonus, not a limitation. We will add guidance in Section 4.2 clarifying this.
> > >
> > > We appreciate your continued engagement and hope this clarifies how our theoretical and empirical contributions complement each other.

---

### Official Review · Reviewer_tyc2 · 2025-10-30

**Soundness:** 2
**Presentation:** 2
**Contribution:** 2
**Rating:** 4
**Confidence:** 3

**Summary:**

The paper proposes modular, adaptive conformal prediction (CP) for two-stage pipelines (and claims it can extend to multi-stage settings). The key idea is to decompose the end-to-end residual into an upstream component (delta R_1) and a downstream residual (R_2), and to construct prediction intervals by combining the component quantiles with scaling weights.

A central issue is that general coverage guarantees do not hold when using arbitrary scaling weights (a, b). To address this, the authors propose selecting (a, b) on the calibration set to achieve a desired coverage target, following the risk-controlling approach from Bates et al. (2021) and Angelopoulos et al. (2021, 2022). Specifically, they apply an FWER-controlled multiple testing procedure over a candidate grid of (a, b).

In addition, the paper proposes an adaptive variant that essentially follows the approach of Gibbs and Candès (2021) to handle non-stationary data by updating the intervals over time. Handling such distribution shifts appears to be the paper's central message and main experimental focus.

**Strengths:**

- The idea of decomposing the end-to-end residual into upstream and downstream components is conceptually appealing and relevant for modular pipelines. The motivation of asymmetric shifts is well argued.
- When a = b = 1, delta R_1 and R_2 are transparently defined and lead to a simple, interpretable coverage guarantee of ≥ 1 − 2α.
- The adaptive mechanism provides a clear way to adjust (a, b, c, d, α) over time, with a long-run coverage statement and stage-aware signals determining which component to update.

**Weaknesses:**

- Section 4 introduces a, b, c, d, but their roles and how to choose them are never explained clearly. Readers only gather that c and d are component-wise quantile levels and a and b are scaling weights. However, there is no practical guidance (e.g., defaults, grids, or heuristics) before moving straight into the risk-control framework.
- After noting that general (a, b) coverage is hard to guarantee, the paper abruptly transitions to the FWER selection, and then directly to distribution shift experiments. Although shift robustness is motivated earlier, the connection between the FWER-based method and robustness to shifts is not clearly explained, creating a disjoint flow.
- While the method is said to generalize to multi-stage pipelines, this seems largely infeasible in practice, as the search space of parameter combinations grows exponentially with the number of stages.
- The approach is overall conservative (wide intervals and abstentions). Although τ is proposed as a way to balance coverage and efficiency, the discussion remains qualitative, leaving practitioners unsure about practical defaults.
- The method introduces many hyperparameters (Λ grid, τ, δ, k, γ, η, c, d), adding complexity. While ablations exist, the main text could more effectively summarize robust or recommended settings

**Questions:**

- What exactly is meant by “separate component quantiles” (what is its purpose), and how should these be chosen in practice?
- In the first non-adaptive experiment under a non-stationary setting, most baselines (except WSC) are not designed for time-varying data, including your own non-adaptive approach. The improved coverage you report seems mainly due to the method’s inherent conservativeness, which other methods can also achieve by using a smaller significance level. Additionally, does the calibration set expand over time or remain fixed? If it is fixed, the WSC baseline loses its main advantage (adaptive weighting).
- In the adaptive experiments, I would like to see a baseline where the upstream stage is removed, and a single model directly predicts y, followed by applying ACI, PID, or OCID. Currently, the setup treats the two-stage pipeline as a black box, making it unclear how much of the benefit comes from the modular decomposition.
- How sensitive are your conclusions to misspecified intermediate representations x (e.g., if x is noisy or partially observed)?

---

> ### Author Response · Authors · 2025-11-20
>
> Thank you for your thoughtful review highlighting important implementation and clarity matters.
> **Hyperparameter Guidelines** We have added a comprehensive set of hyperparameter guidelines in Appendix D.2 which we refer to in Section 4 and 7 and display below.
> - $\Lambda$ Grid: $\\{(a, b) : a, b \in \{0.1, 0.2, 0.3, ..., 1 - \alpha/2\}, \, a + b \geq \frac{1-\alpha}{2} \\}$ This grid can be refined in depending on size of calibration set. Our default was grid of {0.1, 0.2, ..., 1}.
> - Initial/Fixed Values for $c$ and $d$:
>   - Default: $c = d \leq \alpha/2$ (balanced)
>   - Upstream sensitivity: $c = \alpha, d \leq \alpha/2$
>   - Downstream sensitivity: $c \leq \alpha/2, d = \alpha$
>   - Intuition is that $c,d$ control stage-specific sensitivity to shifts, with larger values making that stage more likely to trigger abstention if its error dominates.
> - $\delta$: Default value of 0.3 (moderate); 0.1 (conservative choice with more abstention)
> - Tolerance $\tau$: Default 0 (conservative); $\tau \in [0.03, 0.05]$ improves (Table 5) in "easier" regimes (e.g., IID)
> - Window Size $k$: Recommend $k > 40$ for sufficient data
> - Step Size $\gamma$: Default 0.01; grid of [0.01, 0.03, 0.05, 0.1] should suffice for most cases
> - Step Size $\eta$: Default 0.01 for stability
>
> **Connection between FWER and shift robustness**
> We have changed the writing to make the transitions between $(a,b)$ coverage, FWER selection, and experiments sections smoother and make the connection between FWER and shift robustness. To that end, we have changed the end of Section 4 (line 247-252), the beginning and end of Section 5(line 258-262, line 319-323), and the introduction of Section 7(line 353-354). We also summarize the general points below:
>
> Because general $(a,b)$ weights do not provide coverage guarantees, we adopt a conservative approach to parameter selection using FWER control, ensuring that miscoverage rarely exceeds the threshold $\alpha$ which provides coverage guarantees. FWER's conservatism creates a “safety buffer”—by requiring strong evidence of coverage, we select parameters that remain valid under mild shifts. When shifts exceed this buffer, $\Lambda_{val} = \emptyset$ provides a clear retraining signal, unlike
> black-box methods that silently fail. Thus, FWER-based selection offers a safety margin under distribution shifts, and stage-wise adaptations to parameters (across multiple stages) further improve robustness to shifts. Of course, this conservatism does result in inefficiency on "easier" IID settings which you point out and we address down below with $\tau$.
>
> **Conservativeness**
> It is true that the method is inherently conservative. However, there is nuance; we would like to reiterate the following key features of our approach: (i) Targeted adaptation --- our method adjusts parameters $(a,b)$ based on the dominant source of error, unlike black-box methods that apply a blanket widening approach without considering the source of uncertainty. (ii) Diagnostic abstention -- when $\Lambda_{val} = \emptyset$, the method signals which stage requires retraining rather than producing unreliable intervals. This diagnostic feature helps to pinpoint the source of uncertainty, a capability that cannot be achieved by simply tuning $\alpha$ in black-box methods.
> (iii)
> Tolerance parameter $\tau$ allows us to adjust the conservativeness by tightening intervals (as you noted). We include an experiment: https://imgur.com/a/4o5u7DM which visualizes performance with $\tau=0.05$ in the non-adaptive IID setting across varying $\alpha$. Our method (blue) is more efficient than WSC (purple) and is competitive in efficiency with SC (green) as $\alpha$ decreases (darker colors). We also provide our method with $\tau=0$ as a comparison, which is much more inefficient because it filters away tighter weights (which is unnecessary in IID settings).
>
> To recap: black-box methods with smaller $\alpha$ achieve coverage through blind widening, while ours has accurate component weighting and diagnostic information based on error sources.
>
> **Multi-stage Extension**
> You are correct that the space of parameter combinations grows exponentially with the number of stages. Specifically, if the grid for $\Lambda$ has size $v$, then for $N$ stages, the search space becomes $O(v^N)$, which is infeasible to explore fully.
>
> In Appendix C.4, we discuss restricting $\Lambda$ for practical deployment. We focus on adjusting the top $u$ parameters by magnitude of residual components, freezing the parameters for the remaining stages. This approach is based on intuition that not all stages contribute equally to the error, so focusing on the most impactful ones is reasonable. This reduces the search space to $O(v^u)$. This can be performed sequentially, adjusting one stage at a time, reducing the complexity to $O(vu)$, at the cost of a less exhaustive search. We recognize that more efficient and thorough multi-stage search techniques are an important direction for future work.

---

> ### Author Response · Authors · 2025-11-20
>
> **Single-stage Baseline**
> We thank you for this excellent suggestion. We have done comprehensive experiments for automobile indicators and synthetic with sudden shifts where we apply the adaptive baselines to a direct single-stage $w \to y$ predictive model rather than the two-stage approach.
>
> **Automobile Indicators Results** (Figure in Appendix D.4.5), direct link : https://imgur.com/a/BmnXVlI
>
> We consider two options for the single-stage model, either ARIMA(2,0,1) or a VAR model. We observe:
>
> ACI and PID still fail to capture the 2020 COVID distribution shift. Furthermore, the coverage remains similar between single and two-stage. For example, ACI and PID end with coverage of \~0.53,\~0.55 (ARIMA) or \~0.61,\~0.61 (VAR) respectively, compared to \~0.58,\~0.6 with the two-stage model, still well below desired 0.8. OCID fares better but remains inefficient, with slightly worse performance on single-stage \~0.75 (ARIMA),\~0.71(VAR) compared to \~0.84 on the two-stage model.
>
> **Synthetic Dataset Results** (Figure 18 in Appendix D.4.5, link: https://imgur.com/a/5nPVwcg):
>
> All methods still struggle with the sudden distribution shifts, even with the single least-squares model. Despite slightly tighter intervals on average, they still fail to maintain coverage during shifts.
>
> Thus we have the key findings:
>
> 1. The two-stage pipeline provides predictive benefits over standard single-stage time-series models on this dataset.
> 2. However, baseline models struggle to compensate for stage-wise distribution shifts for both single-stage and two-stage models so the quality of the model doesn't necessarily improve coverage.
> 3. Taken together, the improvements in coverage from our method stem from explicitly leveraging pipeline structure via decomposition, not just from using a better predictive model, as otherwise the other baselines would see similar improvement.
>
> For further intuition, consider the COVID shock. Black-box methods see
> only total error increasing and uniformly widen intervals. Our decomposition uses the fact that $\Delta R_1$ (upstream/supply chain) spikes while $R_2$ (downstream/pricing) remains relatively stable, adapting the weights in a targeted manner. This specific response is impossible without the decomposition.
>
> **Additional Questions**
>
> **Noisy Intermediate Representation**: The decomposition remains functional with noisy/imputed $x$. While distortion affects weights, the impact is minimal when stage imbalance is significant (the dominant stage still emerges clearly), so for small amounts of noise it should work fine. More sophisticated methods for handling noisy or missing data could be explored to improve the robustness of the decomposition.
>
> **Separate Component Quantiles**:  By "separate component quantiles," we mean taking quantiles of $\{\Delta R_1\}$ and $\{R_2\}$ independently using the quantile level parameters $c$ and $d$, then combining. This demonstrates how decomposed residuals can generate intervals with guarantees.
>
> **Non-adaptive Experiment Clarification**: We have addressed the interactions of conservativeness of our method in non-adaptive settings above and how other methods cannot simply decrease $\alpha$ to get the same diagnostic behavior. Further note that many baselines which explicitly deal with time-varying data are adaptive. Also, we have tested CQR as an additional non-adaptive baseline which we consider adding. Lastly, the calibration set is not fixed, so WSC *is* able to weight adaptively.
>
> Your feedback has substantially improved the paper's clarity. The single-stage experiments in particular have helped highlight that our gains come from the decomposition enabling adaptation rather than from improved predictive power. The diagnostic information of our method from the conservative FWER procedure, combined with the option of competitive efficiency via $\tau$ justifies practical deployment over black-box alternatives that fail similarly but provide no structural insights. Thank you for your suggestions.

---

### Official Review · Reviewer_rQHP · 2025-11-01

**Soundness:** 3
**Presentation:** 3
**Contribution:** 3
**Rating:** 4
**Confidence:** 4

**Summary:**

The paper introduces a novel conformal method that decomposes the residual into two stages. For the training pipeline with multiple training stages, the method offers interpretability while preserving long-run coverage guarantees.

**Strengths:**

The idea to decompose the uncertainty into relevant components based on the stages of the training pipeline is interesting and can offer better insight into modelling or fine-tuning under distribution shift.

**Weaknesses:**

The biggest weakness is the comparison with only one significance level. To best judge a conformal method, it is imperative to compare its performance across different significance levels, and plotting a calibration curve is particularly helpful.

See "Questions*" below for more

**Questions:**

1. Line 50: "interpretable intervals": I believe the interpretability here refers to the decomposition. It could be misinterpreted as rectangular regions if the response variable is multivariate.

2. Line 104-116: Different properties, such as Exchangeability and Distribution Shifts, start suddenly; consider numbering them as done later in the paper.

3. Line 137: The term attribution appears odd. I believe the authors wanted to imply decomposition.

4. If I understood correctly, the interval provided in Definition 4 is a direct application of the triangle inequality mentioned earlier. This seems to create the interval that would lead to overcoverage more often than not. Is it also possible to have exact coverage guarantees in such a case?

5. In Proposition 2, there could be multiple sets of (a, b) that satisfy the coverage guarantee; is that correct?

6. One slight issue  I feel in this setup is the efficiency of the conformal prediction, while this decomposition of uncertainty is helpful, if we lose efficiency, it may not be worth.

7. Line 100-101: This is the first time I have heard of a conformal set. Typically, the role that you conformal set has is done by the calibration set. However, it appears you needed an extra split in this case for tuning the parameters. Does it make the method data-inefficient?

8. The long-term guarantee seems to be the same as in Gibs and Candes 2021. Is that correct?

9. Line 372: 'Concept Shift"?  Did you mean 'distribution shift'?

10. Line 406,/423: "Distribution Shocks"/ "upstream shocks"? Did you mean 'distribution/upstream shift' again?

11. The target confidence is set as high as 0.9, which is okay. However, it is necessary to demonstrate the performance of the proposed method with varying significance levels. A calibration curve may be helpful in determining if the proposed method works effectively in all cases.

---

> ### Author Response · Authors · 2025-11-20
>
> Thank you for your detailed review and suggestions. Your main concern about testing across multiple significance levels is valid and has led to improved thoroughness.
>
> ## Main Concern
>
> To address your main concern, we have conducted extensive new experiments varying $\alpha$ levels and plotted calibration curves and width analysis as you recommended.
>
> **Calibration and Width Analysis, Automobile Data (2019-2023)**
> We vary $1- \alpha \in [0.5,0.9]$, comparing our adaptive method against the baselines ACI, PID, OCID and plotting calibration curves and width analysis. We have updated the manuscript to include these results in Appendix D.4.3, Figure 10 and also provide a anonymized link: https://imgur.com/a/XbsUnPT.
>
> In terms of calibration performance, our method tracks the diagonal well between [0.75,0.9), achieving 0.83 coverage when desired is 0.85. Our method is more conservative between [0.5,0.6], achieving more coverage than necessary, however during that period our width is actually second lowest, beating PID and OCID. Furthermore, while OCID is generally the best in coverage, our method displays less erratic behavior and remains tighter in width.
>
> For width, our method obtains competitive efficiency, comparable to ACI and PID and more efficient than OCID. At 0.7 coverage, Our width ~24 vs. ACI's ~22, PID's ~25, OCID's ~32. At 0.85 coverage: Our width ~37 vs. ACI's ~35, PID's ~36, OCID's ~38. At 0.90 coverage: Our width ~40 vs. PID's ~41, ACI's ~37, while OCID
> spikes to ~64.
>
> Note the dip in coverage for our method at 0.90 is due to abstentions, which we count as coverage failures to be more critical, even if it might be a valid output. This abstention signals when uncertainty is too high for reliable intervals—a diagnostic feature unavailable in baseline methods.
>
> **Non-Adaptive IID Analysis** We analyze efficiency across varying $\alpha \in[0.05,0.15]$ (indicated by shade-darker = smaller $\alpha$). We use the tolerance parameter $\tau=0.05$. This is Figure 4 in Appendix D.3.2. We provide a direct link as well: https://imgur.com/a/4o5u7DM.
>
> Noticeably, at smaller values of $\alpha$, our method (blue) approaches and overlaps with SC's (green) performance, achieving coverage ~0.95 with width 2.4, the same as SC; in comparison WSC (purple) has ~0.96 with width 2.5 for desired coverage level $0.95$. Notably, WSC consistently maintains a gap from SC. The tolerance parameter $\tau$ is crucial for this setting, as we provide a comparison for our method with 0 tolerance (orange), which overcovers significantly.
>
> This demonstrates our method can match SC in efficiency under IID settings across multiple $\alpha$ using the tolerance parameter while still providing stage-wise weights for diagnostics.
>
> **Responses to Specific Questions**
> - Q1,Q3,Q9,Q10: **Terminology clarifications:** By "interpretability" and "attribution," we refer to error decomposition across stages. "Attribution" specifically means identifying which stage contributes most to the overall error (as defined in line 126). We've changed interpretability to stage-wise error decomposition. distribution shift terms: "Concept shift" (defined in line 118) refers to a stagewise shift, where the relationship between $w$ and $x$ or $x$ and $y$ changes. We’ve replaced "concept shift" with "upstream" or "downstream" shift for consistency. "Shocks" referred to sudden, drastic shifts, but we agree it's imprecise. We've updated the manuscript to use "sudden distribution shift" instead.
> - Q2: We have numbered the assumptions in lines 104-116 as suggested.
> - Q4: Correct, Definition 4 with $a=b=1$ is a conservative upper-bound via triangle inequality. However Definition 4 allows for other weight choices that are not directly an upper-bound. This is exactly \textit{why} we state Proposition 2, as it shows exact coverage is theoretically achievable; FWER empirically approximates optimal weights.
> - Q5: Yes, there can be many such $(a ^ *, b ^ *)$. Given the weights are in [0,1] this set won't have excessively large range. This is analogous to the multiple options in $\Lambda_{val}$ when selecting the $(a,b)$.
> - Q6: Our new results above address efficiency concerns: we match SC efficiency across varying $\alpha$ and have competitive performance compared to adaptive baselines for non-adaptive IID.
> - Q7: The "conformal set" is introduced to avoid contaminating the calibration set with data used for generating the residual components.
> We use the same total held-out data as baselines, splitting it in two to form $S^{conf}, S^{cal}$. Empirically $S^{cal}$ has been as small as 36 points. We do an ablation study on the sliding-window size in Appendix D.4.4, table 11.
> - Q8: Yes, we extend Gibbs and Candès framework to our augmented parameter space.
> - Q11: Addressed in main concern.
>
> Your request for more significance levels reveals our method's strengths persist across multiple values of $\alpha$.
> Thank you again for pushing us to provide this more complete evaluation.

---

> > ### Comment · Reviewer_rQHP · 2025-11-28
> >
> > Thanks for writing the rebuttal and for the clarifications.
> >
> > While there are significance levels in [0.5,0.9], I wish there were more levels there. Some conformal methods show better coverage when low significance levels are chosen, but not when they are higher. The stability of the method can be best assessed at higher levels, and hence I had asked for a calibration plot.
> >
> > Nonetheless, my understanding is that the method not only undercovers at 0.1 significance but at other levels as well (such as 0.25). Could the authors comment on it?
> >
> > "Note the dip in coverage for our method at 0.90 is due to abstentions, which we count as coverage failures to be more critical, even if it might be a valid output. This abstention signals when uncertainty is too high for reliable intervals—a diagnostic feature unavailable in baseline methods." - could the authors elaborate more on it as well?

---

> > > ### Author Response · Authors · 2025-11-29
> > >
> > > We had initially considered $[0.5, 0.9]$ sufficient, as values below $0.5$ are rarely used in practice; however, we now understand the reviewer’s intent. We have rerun the experiment to include $\alpha$ values across $[0,1]$ and have included the results in this anonymous link: https://imgur.com/a/asIvGGm.
> > >
> > > We acknowledge the slight undercoverage at certain $\alpha$ values, but we note that the other baselines (excluding OCID) also exhibit similar undercoverage, as no method guarantees exact coverage when exchangeability is violated. Thus, occasional deviations are expected in finite samples, as empirical quantiles change discretely with $\alpha$ and are not necessarily monotonic. We would also like to emphasize that our method maintains stronger coverage relative to ACI and PID, remaining overall closer to or above the desired threshold across the range of $\alpha$, while OCID achieves the best coverage but produces substantially wider intervals.
> > >
> > > Regarding the dip at $\alpha = 0.9$, this is due to abstention. To be extra conservative in evaluating our method, we count abstentions as miscoverage, which results in the observed dip. Alternatively, one could omit abstentions entirely and report coverage over the remaining points, in which case the coverage is much closer to the desired level (see the third plot in the above link).
> > >
> > > Furthermore, recall that the evaluation period includes the COVID-induced 2020 shock, during which the automobile indicator experienced sudden jumps. This contributes to undercoverage across most baseline methods and reflects the difficulty of the time period. In such a regime of high uncertainty, abstention is a calibrated signal that our method cannot produce a reliable interval for $1-\alpha = 0.9$. By contrast, the other methods always produce an interval, even if unreliable. Thus, abstention is a safety mechanism rather than a failure. Although OCID does achieve the desired coverage even under this difficult setting, its intervals are consistently much wider across all $\alpha$ values and it lacks diagnostic ability, unlike our method, which remains more efficient while providing a principled abstention mechanism.

---

### Author Response · Authors · 2025-11-20
**To All Reviewers:**

Thank you all for your thoughtful and detailed reviews.

We greatly appreciate the time and effort you dedicated to evaluating our work. Your feedback has been instrumental in improving the clarity and quality of the paper.
We are pleased to see that all of you recognized the novelty and usefulness of the residual decomposition method we propose. We are particularly glad that you highlighted its diagnostic ability to identify stage-wise performance, as well as its enhanced coverage compared to traditional black-box conformal methods. Your feedback strengthens our confidence in the contributions of this work.

Once again, thank you for your valuable insights and for helping us improve our manuscript.

---

### Note · Program_Chairs · 2026-01-17
**Submission Desk Rejected by Program Chairs**

The following references in this submission do not refer to real documents and/or have major errors in bibliographic information:

 Carlo Emilio Bonferroni. On the multiple hypotheses test, 1937. origin of the Bonferroni correction